# Weakly-supervised Audio Separation via Bi-modal Semantic Similarity

**Tanvir Mahmud**[1][†][*]**, Saeed Amizadeh**[2][†]**, Kazuhito Koishida**[2] **& Diana Marculescu**[1]
[1]The University of Texas at Austin, USA, [2]Microsoft Corporation
{tanvirmahmud, dianam}@utexas.edu, {saamizad, kazukoi}@microsoft.com

## Abstract

Conditional sound separation in multi-source audio mixtures without having access to single source sound data during training is a long standing challenge. Existing *mix-and-separate* based methods suffer from significant performance drop with multi-source training mixtures due to the lack of supervision signal for single source separation cases during training. However, in the case of language-conditional audio separation, we do have access to corresponding text descriptions for each audio mixture in our training data, which can be seen as (rough) representations of the audio samples in the language modality. That raises the curious question of how to generate supervision signal for single-source audio extraction by leveraging the fact that single-source sounding language entities can be easily extracted from the text description. To this end, in this paper, we propose a generic bi-modal separation framework which can enhance the existing unsupervised frameworks to separate single-source signals in a target modality (*i.e.*, audio) using the easily separable corresponding signals in the conditioning modality (*i.e.*, language), *without having access to single-source samples in the target modality during training*. We empirically show that this is well within reach if we have access to a pretrained joint embedding model between the two modalities (*i.e.*, CLAP). Furthermore, we propose to incorporate our framework into two fundamental scenarios to enhance separation performance. First, we show that our proposed methodology significantly improves the performance of purely unsupervised baselines by reducing the distribution shift between training and test samples. In particular, we show that our framework can achieve 71% boost in terms of Signal-to-Distortion Ratio (SDR) over the baseline, reaching 97.5% of the supervised learning performance. Second, we show that we can further improve the performance of the supervised learning itself by 17% if we augment it by our proposed weakly-supervised framework. Our framework achieves this by making large corpora of unsupervised data available to the supervised learning model as well as utilizing a natural, robust regularization mechanism through weak supervision from the language modality, and hence enabling a powerful semi-supervised framework for audio separation. Code is released at https://github.com/microsoft/BiModalAudioSeparation/.

## 1 Introduction

Environmental sounds often appear in mixtures containing diverse sounding events from different sources (Kim et al., 2019). Separating sounds from mixtures has significant applications in speech (Liu & Wang, 2018), audio (Chen et al., 2022), and music processing (Stöter et al., 2019). Humans gather the perception of every single sounding source through experience and interactions with environment. To train sound separation models, most prior works require well-curated, single source samples (Yu et al., 2017; Sawata et al., 2021). However, procuring large-scale single source datasets for all possible sounding entities can be quite costly and time-consuming in many real scenarios (Wisdom et al., 2020; Dong et al., 2022). Nevertheless, classical unsupervised attempts to separate all sounding sources in a mixture typically require complex post-processing to

---

[*]Work done in part during an internship at Microsoft Corporation, Redmond, USA

[†]Equal contribution.

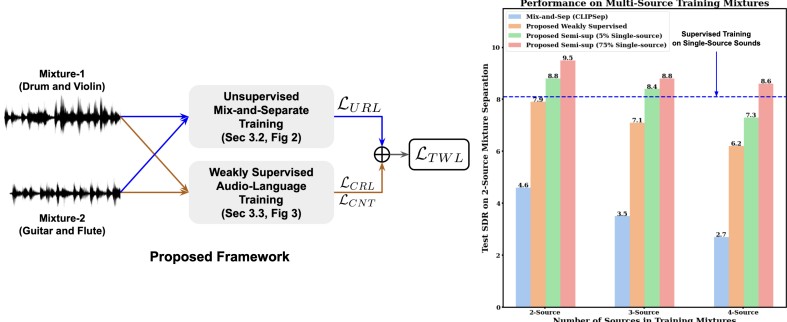

Figure 1: **(Left)** The proposed conditional audio separation framework. **(Right)** The comparison of our framework and the *mix-and-separate* baseline in unsupervised and semi-supervised settings.

isolate the target source which limits their applicability (Wisdom et al., 2020). In contrast, conditional sound separation directly separates the target sound from the input mixture via conditioning signals possibly presented in another modality. Prior work introduced image (Gao & Grauman, 2019), video (Zhao et al., 2018), reference audio (Chen et al., 2022), and language conditions (Dong et al., 2022) to represent the sounds to be separated. Most of these works are built upon the idea of *mix-and-separate*, originally introduced by Zhao et al. (2018).

Despite the success of mix-and-separate approach, the model does not see single-source cases during training which means that the single-source conditioning signals are not seen at the training time either. This creates a shift between training and test distributions, which, in turn, harms the model's generalization. The problem is further exacerbated by the increasing number of components in the training mixtures, because now the model has to indirectly discover single-source patterns from even more complex training patterns. For instance, as shown in Figure 1, the performance of mix-and-separate method on 2-component test mixtures drops below 35% of that of the supervised approach when the number of components in training data increases to four.

Aside from mix-and-separate, incorporating sound classification has been proposed to generate weak supervision for single source samples during unsupervised training (Tzinis et al., 2020; Pishdadian et al., 2020). However, such methods suffer from fixed and limited number of single-source sounding entities (*i.e.*, the number of classes) as well as dependency on seeing prototypical, single source examples for each class during training, which severely restricts their applicability in real world scenarios. In a similar vein, in this paper, we seek a methodology to generate weak supervision, but *without* having to deal with the restrictions of the classification-based methods. To this end, we propose a weakly-supervised audio separation framework that relies on incorporating pretrained audio-language embedding models (specifically CLAP (Wu et al., 2023)) into the unsupervised learning training loop. Since models such as CLAP (i) show promising open-ended recognition and zero-shot capabilities, and (ii) are already pretrained on large corpora of audio-language pairs, our proposed framework breaks the classification-based methods' barrier while at the same time (unlike mix-and-separate) provides weak supervision for single-source audio separation during training. More generally, the main contributions of our work are:

(1) We propose a weakly supervised source separation framework capable of extracting single-source signals from a mixture in a target modality (*e.g.*, audio) using a conditioning modality (*e.g.*, language) if (i) we can easily separate the corresponding entities in the conditioning modality, and (ii) we have access to a pretrained joint embedding model between the two modalities. In this paper, we adapt this general framework for language-conditional audio separation.

(2) We incorporate our framework into a pure unsupervised setting and show that, compared to mix-and-separate, our method achieves up to 71%, 102% and 129% SDR boost on 2-component separation test when trained on 2-, 3-, and 4-component mixture data, respectively (Figure 1).

(3) We further propose to use our methodology to augment supervised learning-based audio separation into a powerful semi-supervised learning (SSL) counterpart. We show that in moderate SSL scenarios, we achieve 17%, 8%, and 6% SDR boost over supervised learning for 2-component separation when trained on 2-, 3-, and 4-component mixture data, respectively; whereas for extreme SSL scenarios, our method still outperforms the supervised training by 8% and 3% when trained on 2- and 3-component mixture data, respectively. (Figure 1).

(4) We conduct extensive experiments and ablation studies on MUSIC (Zhao et al., 2018), VG-GSound (Chen et al., 2020), and AudioCaps (Kim et al., 2019) datasets, and report the results.

## 2    RELATED WORKS

**Unconditional Sound Separation**    Most prior work focused on unconditional sound separation on speech (Wang & Chen, 2018; Yu et al., 2017; Zhang et al., 2021; Luo & Mesgarani, 2018), and music (Stöter et al., 2019; Sawata et al., 2021; Takahashi & Mitsufuji, 2021). For these methods, post-processing mechanisms have been employed to pick the target sound. Kavalerov et al. (2019) used permutation invariant training (PIT) for universal sound separation, originally introduced by Yu et al. (2017). PIT measures permutations of all predictions to match the ground truth sounds. Nevertheless, these methods still need single source samples for training. Later, Wisdom et al. (2020) proposed a mixture invariant training (MixIT) to use multi-source mixtures during training, but suffers from over-separation of sounding sources, and still requires post-processing to separate the sounds. Later, Wisdom et al. (2021) proposed to use a pretrained sound-classifier to separate the sounds while training. Such a classifier, however, requires single-source samples for training. MixPIT (Karamatlı & Kırbız, 2022) proposed direct predictions of mixtures from mixture of mixtures (MoM) in training but it suffers from under-separation. Pishdadian et al. (2020) proposed a weakly supervised training by applying sound classifiers on estimated sources; however, the model is restricted by the fixed number of classes. In contrast, our framework achieves single-source audio separation with open-ended natural language prompts, *without* relying on any post-processing or having access to single-source samples during training.

**Conditional Sound Separation**    Prior work on conditional sound separation utilized visual information (Gao & Grauman, 2019; Zhao et al., 2018; Tian et al., 2021; Chatterjee et al., 2021; Lu et al., 2018), as well as text information (Dong et al., 2022; Liu et al., 2022; Kilgour et al., 2022; Tan et al., 2023; Liu et al., 2023) of sources in mixtures for separation. Most of these methods employ the mix-and-separate framework originally introduced by Zhao et al. (2018). Recently, Dong et al. (2022); Tan et al. (2023) introduced modality inversion conditioning by leveraging CLIP model with mix-and-separate, where video/image conditioning is used for training while both video and text can be used for test conditioning. However, with increasing number of sources in training mixtures, the performance of these methods significantly drop compared to supervised training. Gao & Grauman (2019) introduced a sound classifier in mix-and-separate for clustering single source samples. AudioScope (Tzinis et al., 2020) added a classifier in the MixIT baseline for separating on-screen samples based on visual conditioning. Another line of research uses reference audio signal for conditioning in mixture separation (Chen et al., 2022; Gfeller et al., 2021). However, it is generally costly to gather reference signals for single sources. In contrast, our approach can provide competitive results of single source training under completely unsupervised scenarios.

**Contrastive Learning on Audio, Language, and Vision Foundation Models**    Radford et al. (2021) first introduced CLIP model which learns joint visual-language embedding from 400M image-text pairs. CLIP has been extensively studied in diverse visual-language applications, such as zero-shot classification (Radford et al., 2021), open-vocabulary segmentation (Luo et al., 2023; Xu et al., 2023), and object detection (Zareian et al., 2021; Du et al., 2022). Guzhov et al. (2022) later integrates audio modality to learn joint representations of audio, vision, and language. Wu et al. (2023) introduced joint audio-language embedding with CLAP model by large-scale pretraining with $633, 526$ audio-caption pairs. In this work, we leverage pretrained CLAP model to generate weak supervision with representative texts in multi-source sound separation training.

## 3    THE PROPOSED FRAMEWORK

We propose a robust language-conditional sound source separation framework capable of separating single sources from mixtures *without having access to single-source audio data during training*. To this end, we develop a generic weakly-supervised training framework to separate/segment a single-source signal from a mixture in modality $\mathcal{A}$ conditioned on a single-source, concurrent signal in modality $\mathcal{T}$ assuming that we have access to (i) single-source signals in modality $\mathcal{T}$, and (ii) a pretrained joint embedding model between modalities $\mathcal{A}$ and $\mathcal{T}$. If these requirements are met, our framework can learn a conditional source separation model in modality $\mathcal{A}$ without needing single-source instances from this modality during training. In this sense, our proposed framework is *weakly-supervised*. In this paper, we focus on the specific scenario where modality $\mathcal{A}$ is audio and modality $\mathcal{T}$ is language. As such, requirement (i) boils down to having access to single-source entities in language (*e.g.*, "the sound of violin") which is easily obtainable from the caption associated with an audio mixture sample using standard NLP techniques such as Name Entity Recognition

(NER) (Li et al., 2020) combined with basic prompt engineering. Also requirement (ii) is met as we do have access to language-audio pretrained models such as CLAP (Wu et al., 2023).

Nevertheless, driving training purely based on weak supervision will only get us so far. In practice, pure weak supervision through language is quite often coarse and therefore produces low fidelity audio samples during inference. To overcome this problem, the conventional wisdom is to incorporate some form of *reconstruction loss* minimization during training. However, traditionally, reconstruction loss relies on having access to supervised training data (*i.e.*, single-source audio samples) which we are trying to circumvent. As a result, here we propose to incorporate the unsupervised version of the reconstruction loss minimization, aka the *mix-and-separate* approach (Zhao et al., 2018) in combination with our weakly-supervised training, as depicted in Figure 1.

## 3.1 PROBLEM FORMULATION

Let $\mathcal{D} = \{(\mathcal{M}_1, \mathcal{T}_1), (\mathcal{M}_2, \mathcal{T}_2), \ldots, (\mathcal{M}_P, \mathcal{T}_P)\}$ be a set of $P$ audio mixture and text description pairs. We assume each audio mixture $\mathcal{M}_i$ is composed of $K > 1$ single source audio sounds $\mathcal{S}_i^k$, *i.e.* $\mathcal{M}_i = \sum_{k=1}^{K} \mathcal{S}_i^k$, which we do *not* have access to during training. Furthermore, we assume each audio single source $\mathcal{S}_i^k$ corresponds to a *sounding language entity* $\mathcal{T}_i^k \in \mathcal{T}_i$, which is either given or easily obtainable from the textual prompts $\mathcal{T}_i$. For example, for a music audio mixture accompanying with the textual description "a trio of violin, piano, and cello", the sounding language entities are "violin","piano", and "cello". Given this setting, our goal is to train a neural network model $f_\theta$ using $\mathcal{D}$ such that, at inference time, given a mixture $\mathcal{M}$ and a sounding language entity $\mathcal{T}^k$, it extracts the corresponding single-source audio $\mathcal{S}^k = f_\theta(\mathcal{M}, \mathcal{T}^k)$ from $\mathcal{M}$[1]. More precisely, we design $f_\theta$ as a residual conditional U-Net that operates on the magnitude spectrogram of the input audio mixture:

$$f_\theta(\mathcal{M}, \mathcal{T}^k) = S^{-1}\big[|S(\mathcal{M})| \odot g_\theta(|S(\mathcal{M})|, \varepsilon_L(\mathcal{T}^k)), \phi\big(S(\mathcal{M})\big)\big] \tag{1}$$

where $S(\cdot)$ is the Short Term Fourier Transform (STFT) function, $|\cdot|$ and $\phi(\cdot)$ are the magnitude and phase functions, $g_\theta$ is the masking U-Net function parameterized by $\theta$, $\varepsilon_L(\cdot)$ is the (frozen) language embedding model, and $\odot$ is the Hadamard product. Note that our design choices for modeling $f_\theta$ are specifically guided by the nature of the problem at hand. First, our U-Net operates only on the magnitude of the input mixture as it is known that the phase information is not crucial to many audio prediction tasks. Second, we have intentionally chosen the mask architecture for $f_\theta$ as opposed to directly predicting the output in order to encode the inductive bias that the output of $f_\theta$ should be a component of its input mixture. This inductive bias is crucial for the case of weakly supervised learning, as the supervision signal does *not* contain the fine-grained information necessary for reconstructing the output from scratch.

The masking function $g_\theta$ is a conditional U-Net that predicts a (soft) binary mask for an input magnitude spectrogram mixture conditioned on the encoding of the text prompt condition (via $\varepsilon_L(\cdot)$). We have extended the common audio U-Net architecture to incorporate the conditioning signal at different resolutions of the input by using multi-scale cross attention modules. For more details of the proposed architecture and our architecture ablation study, see Appendices C, and G.1.

## 3.2 UNSUPERVISED MIX-AND-SEPARATE TRAINING

A common methodology to train the neural model presented in equation 1 is to synthetically mix two or more single-source audio samples at the training time and have the model predict each component by incorporating the $\ell_1$ reconstruction loss on the prediction of the model and the original single-source ground truth components. However, in our setting, we do not have access to single-source audio samples during training; instead, we can mix two or more multi-source mixtures and have the model separate them. This is the essence of the unsupervised *Mix-and-Separate* approach. In particular, during training, we synthetically mix two or more audio mixtures to form a *mixture of mixtures (MoM)* which is then fed to the model to separate the original ingredient mixtures.

For example, suppose we have two audio mixtures $\mathcal{M}_1$, with description $\mathcal{T}_1$ as *"a duet of piano and cello"*, and $\mathcal{M}_2$ with description $\mathcal{T}_2$ as *"a duet of guitar and violin"*. We can mix $\mathcal{M}_1$ and $\mathcal{M}_2$ to produce the MoM $\mathcal{M} = \mathcal{M}_1 + \mathcal{M}_2$, which is fed to the model conditioned by either of $\mathcal{T}_1$ or $\mathcal{T}_2$ to estimate $\mathcal{M}_1$ or $\mathcal{M}_2$, respectively. Then the *unsupervised reconstruction loss (URL)* for $\mathcal{M}$ is:

$$\mathcal{L}_{URL}(\mathcal{M}; \theta) = \|f_\theta(\mathcal{M}, \mathcal{T}_1) - \mathcal{M}_1\|_{\ell_1} + \|f_\theta(\mathcal{M}, \mathcal{T}_2) - \mathcal{M}_2\|_{\ell_1} \tag{2}$$

---

[1]Note that in practice, the sounding entity is presented as a prompt phrase to the model, *e.g.*, "the sound of violin" instead of just "violin". For more details of our prompt engineering pipeline, see Appendices G.8 & E

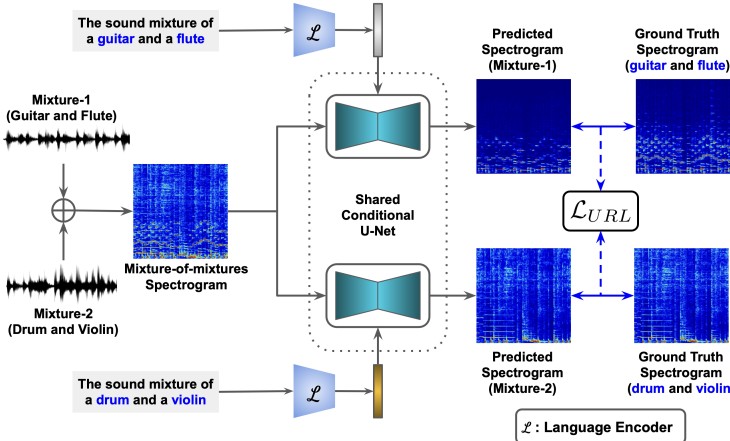

Figure 2: Unsupervised *mix-and-separate* training with language conditioning (for $N = 2, K = 2$)

The generalization of this loss function to more than two components is then straightforward.

### 3.3 WEAKLY SUPERVISED AUDIO-LANGUAGE TRAINING

Despite its merits, the mix-and-separate approach presented in the previous section suffers from a major drawback: due to our unsupervised restriction, the model never sees any single-source conditioning prompt during training; whereas, during inference, most realistic scenarios involve querying for separation of a single-source sound from the mixture input. This creates a distribution shift between the training and testing samples which significantly harms the model generalization. Therefore, the main question is, if we introduce single-source text prompts during training, how can we generate supervision *without* requiring to see the corresponding single-source audio samples?

Our key idea in response to this question is to use the single-source text prompts themselves for supervision! In order to achieve this, we note that the conditioning text prompts and the model's predictions belong to two different modalities (*i.e.*, language and audio) which means that in order to define any notion of similarity between them (which is needed to define a loss function), we would need to map both to a common semantic space. Fortunately, this is a solved problem already with the recent rise of joint multi-modal embedding frameworks such as CLIP (Radford et al., 2021) and CLAP (Wu et al., 2023). In particular, we propose to use the pretrained CLAP language encoder (denoted by $\varepsilon_L(\cdot)$) and the pretrained CLAP audio encoder (denoted by $\varepsilon_A(\cdot)$) to calculate cosine similarity between the the model's prediction and the conditioning text prompts and subsequently generate weak supervision signal for single-source text prompts. However, directly using (negative) cosine similarity as the loss function can result in degenerate outputs, which leads to incorporating cosine similarity within a discriminative setting, *i.e.*, the contrastive loss (Radford et al., 2021). In particular, given a batch of $N$ mixture and text description pairs $\mathcal{B} = \{(\mathcal{M}_1, \mathcal{T}_1), \dots, (\mathcal{M}_N, \mathcal{T}_N)\}$, if each text description $\mathcal{T}_i$ consists of $K_i$ single-source sounding language entities $\mathcal{T}_i^k, k \in 1..K_i$ (*e.g.*, "the duet of piano and cello" consists of "piano" and "cello" single-source sounding language entities), then our flavor of contrastive loss for mixture $\mathcal{M}_i$ is defined as:

$$\mathcal{L}_{CNT}(\mathcal{M}_i; \theta) = -\frac{1}{2K_i} \sum_{k=1}^{K_i} \big( \log \big[ \zeta_\tau(c_{ikik}; \{c_{iajb} : a \in 1..K_i, j \in 1..N, b \in 1..K_j\}) \big] +$$
$$\log \big[ \zeta_\tau(c_{ikik}; \{c_{jbia} : a \in 1..K_i, j \in 1..N, b \in 1..K_j\}) \big] \big) \quad (3)$$

where

$$\zeta_\tau(x; Y) = \frac{\exp(x/\tau)}{\sum_{y \in Y} \exp(y/\tau)}, \text{ and } c_{ikjt} = \frac{\varepsilon_A\big(f_\theta(\mathcal{M}_i, \mathcal{T}_i^k)\big) \cdot \varepsilon_L\big(\mathcal{T}_j^t\big)}{\|\varepsilon_A\big(f_\theta(\mathcal{M}_i, \mathcal{T}_i^k)\big)\|_2 \|\varepsilon_L\big(\mathcal{T}_j^t\big)\|_2} \quad (4)$$

are the Softmax function with (learnable) temperature parameter $\tau$ and the cosine similarity, respectively. If $\mathcal{M}_i$ is a MoM with $M$ component mixtures (that is, $\mathcal{M}_i = \sum_{j=1}^{M} \mathcal{M}_{ij}$), then we define $\mathcal{L}_{CNT}(\mathcal{M}_i, \theta) = \sum_{j=1}^{M} \mathcal{L}_{CNT}(\mathcal{M}_{ij}, \theta)$. We emphasize that the weak supervision signal generated as above does not contain enough information to reconstruct the fine-grain details of the queried single-source audio signal; in fact, the contrastive loss in equation 3 only enforces the predicted audio to contain the *"characteristic features"* such that it can be encoded close to the corresponding text prompt in the CLAP semantic space. That is why we refer to this process as "weak" supervision and why it is crucial to use this loss function combined with the unsupervised reconstruction loss in

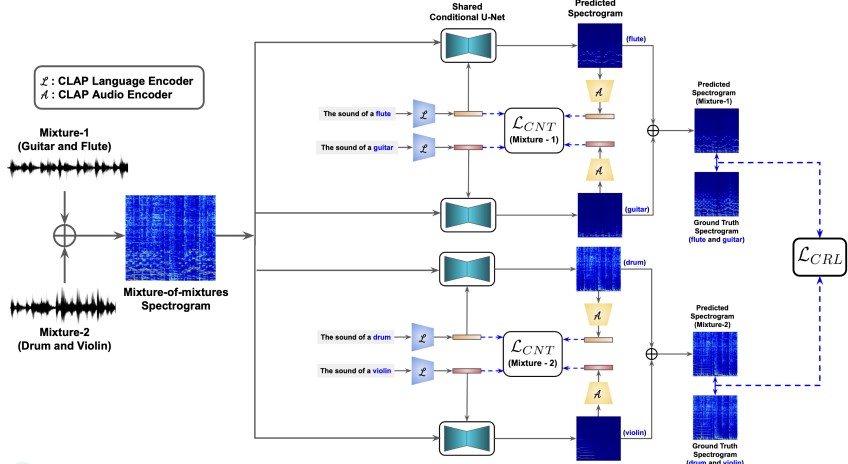

Figure 3: Our proposed weakly-supervised audio-language training framework: bi-modal contrastive loss ($\mathcal{L}_{CNT}$) combined with consistency reconstruction loss ($\mathcal{L}_{CRL}$).

equation 2, which requires us to use MoMs instead of the original mixtures during training. Moreover, since now we can query for single-source audio components, we can add an additional (soft) consistency constraint to have the predicted single source samples sum up to the original mixture. More precisely, for a MoM $\mathcal{M}_i = \sum_{j=1}^{M} \mathcal{M}_{ij}$ where each component mixture $\mathcal{M}_{ij}$ itself consists of $K_{ij}$ (unknown) single-source component $\mathcal{S}_{ij}^k$ for which we have access to the sounding language entities $\mathcal{T}_{ij}^k$, we define the *consistency reconstruction loss (CRL)* as:

$$\mathcal{L}_{CRL}(\mathcal{M}_i; \theta) = \sum_{j=1}^{M} \|\mathcal{M}_{ij} - \sum_{k=1}^{K_{ij}} f_\theta(\mathcal{M}_{ij}, \mathcal{T}_{ij}^k)\|_{\ell_1} \tag{5}$$

Putting everything together, at each training step, given a batch of audio mixtures and their language descriptions $\mathcal{B} = \{(\mathcal{M}_1, \mathcal{T}_1), \ldots, (\mathcal{M}_N, \mathcal{T}_N)\}$, first we randomly combine pairs of mixtures and their text prompts from $\mathcal{B}$ to form the synthetic batch of MoMs $\mathcal{B}' = \{(\mathcal{M}_1', \mathcal{T}_1'), \ldots, (\mathcal{M}_N', \mathcal{T}_{N'}')\}$. Once we have $\mathcal{B}'$, we minimize the *Total Weak-supervision Loss (TWL)* to find the optimal parameter values for our segmentation model $f_\theta$:

$$\mathcal{L}_{TWL}(\mathcal{B}', \theta) = \frac{1}{N'} \sum_{i=1}^{N'} \left[ \alpha \cdot \mathcal{L}_{CNT}(\mathcal{M}_i', \theta) + \beta \cdot \mathcal{L}_{CRL}(\mathcal{M}_i', \theta) + \gamma \cdot \mathcal{L}_{URL}(\mathcal{M}_i', \theta) \right] \tag{6}$$

where $\alpha$, $\beta$ and $\gamma$ are the scalar relative weights for each loss term. Note that the parameters of the CLAP encoder models are kept frozen during training.

### 3.4 SEMI-SUPERVISED LEARNING

As mentioned earlier, the proposed learning framework not only boosts the performance of unsupervised audio separation by introducing weak supervision, but also improves the performance of supervised audio separation by incorporating potentially large corpora of mixture audio data which are not naturally accessible to supervised learning methods. The latter scenario effectively presents a semi-supervised learning (SSL) scheme which can be significantly impactful in practice, as mixture audio datasets are generally way more abundant than their well-curated single source counterparts. Moreover, as we will later see in our experiments, the introduction of unsupervised samples in the supervised training loop via the weak supervision signal provides an elegant regularization mechanism which protects the supervised method from potential overfitting.

In the SSL scenario, in addition to the mixture data, we also have a portion of single-source audio samples in each batch, $\mathcal{S}$, from which we can generate the synthetic set of mixtures $\mathcal{S}'$. Then we simply augment our total loss in equation 6 to include the standard reconstruction loss for $\mathcal{S}'$:

$$\mathcal{L}_{SSL}(\mathcal{B}' \cup \mathcal{S}', \theta) = \lambda_s \cdot \mathcal{L}_{URL}(\mathcal{S}', \theta) + \lambda_u \cdot \mathcal{L}_{TWL}(\mathcal{B}', \theta) \tag{7}$$

where $\lambda_s$ and $\lambda_u$ are the relative weights. Note that, as opposed to the unsupervised case, the $\mathcal{L}_{URL}$ here is computed over $\mathcal{S}'$ for which we have access to the ground-truth single sources (*i.e.*, $\mathcal{S}$).

## 4 EXPERIMENTS

**Datasets**: We experiment on synthetic mixtures produced from single source MUSIC (Zhao et al., 2018) and VGGSound (Chen et al., 2020) datasets by mixing samples from $n$ sources. We use the

Table 1: Comparison on MUSIC Dataset under the unsupervised setup. The supervised column is also provided as an upperbound. SDR on 2-Source separation test set is reported for all cases. All methods are reproduced under the same setting. * denotes implementation with our improved U-Net model. **Bold** and blue represents the best and second best performance in each group, respectively.

| Method | Single-Source (Supervised) | Multi-Source (Unsupervised) | | |
|---|---|---|---|---|
| | | 2-Source | 3-Source | 4-Source |
| **Unconditional** | | | | |
| PIT* (Yu et al., 2017) | $8.0 \pm 0.26$ | - | - | - |
| MixIT (Wisdom et al., 2020) | - | $3.2 \pm 0.34$ | $2.3 \pm 0.57$ | $1.4 \pm 0.35$ |
| MixPIT (Karamatlı & Kırbız, 2022) | - | $3.6 \pm 0.46$ | $2.1 \pm 0.41$ | $1.7 \pm 0.35$ |
| **Image Conditional** | | | | |
| CLIPSep-Img (Dong et al., 2022) | $6.8 \pm 0.25$ | $3.8 \pm 0.27$ | $2.9 \pm 0.35$ | $2.1 \pm 0.32$ |
| CLIPSep-Img* (Dong et al., 2022) | $7.4 \pm 0.22$ | $4.6 \pm 0.31$ | $3.8 \pm 0.28$ | $2.9 \pm 0.43$ |
| CoSep* (Gao & Grauman, 2019) | $7.9 \pm 0.28$ | $4.9 \pm 0.37$ | $4.0 \pm 0.29$ | $3.1 \pm 0.36$ |
| SOP* (Zhao et al., 2018) | $6.5 \pm 0.23$ | $4.1 \pm 0.41$ | $3.5 \pm 0.26$ | $2.7 \pm 0.42$ |
| **Language Conditional** | | | | |
| CLIPSep-Text (Dong et al., 2022) | $7.7 \pm 0.21$ | $4.6 \pm 0.35$ | $3.5 \pm 0.27$ | $2.7 \pm 0.45$ |
| CLIPSep-Text* (Dong et al., 2022) | $\mathbf{8.3} \pm 0.27$ | $5.4 \pm 0.41$ | $4.7 \pm 0.32$ | $3.8 \pm 0.28$ |
| BertSep* | $7.9 \pm 0.27$ | $5.3 \pm 0.31$ | $4.0 \pm 0.22$ | $3.1 \pm 0.27$ |
| CLAPSep* | $8.1 \pm 0.31$ | $5.5 \pm 0.36$ | $4.3 \pm 0.28$ | $3.5 \pm 0.33$ |
| LASS-Net (Liu et al., 2022) | $7.8 \pm 0.25$ | $5.2 \pm 0.26$ | $4.2 \pm 0.29$ | $3.6 \pm 0.36$ |
| Weak-Sup (Pishdadian et al., 2020) | - | $3.1 \pm 0.47$ | $2.2 \pm 0.38$ | $1.9 \pm 0.33$ |
| Proposed (w/ Timbre Classifier - concurrent training) | - | $5.0 \pm 0.29$ | $4.5 \pm 0.32$ | $3.5 \pm 0.27$ |
| Proposed (w/ Timbre Classifier - pretrained) | - | $6.1 \pm 0.33$ | $5.2 \pm 0.37$ | $4.1 \pm 0.35$ |
| **Proposed (w/ Bi-modal CLAP)** | - | $\mathbf{7.9} \pm 0.35$ | $\mathbf{7.1} \pm 0.42$ | $\mathbf{6.2} \pm 0.38$ |

same test set containing samples of 2 sources for each dataset in all experiments. We also experiment with AudioCaps (Kim et al., 2019), a natural mixture dataset containing $1 \sim 6$ sounding sources in each mixture with full-length captions, where we use Constituent-Tree language parser (Halvani, 2023) to extract single source text phrases. See Appendix E for more details on dataset preparation.

**Training**: All the models are trained for $50$ epochs with initial learning rate of $0.001$. The learning rate drops by a factor of $0.1$ after every $15$ epochs. Adam optimizer (Kingma & Ba, 2014) is used with $\beta_1 = 0.9$, $\beta_2 = 0.999$ and $\epsilon = 10^{-8}$. The training was carried out with $8$ RTX-A6000 GPUs with $48$GB memory. For more implementation and training details, see Appendix D.

**Evaluation**: We primarily use signal-to-distortion ratio (SDR) (Vincent et al., 2006) for evaluating different models. However, we have also calculated other evaluation metrics for comparisons, the details of which can be found in Appendices F and G.3.

**Setup**: Every instance of training in our experiments is carried out under one of these three settings: **(I) supervised**, where the training data consists of only single-source samples, **(II) unsupervised**, where the training data consists of only multi-source samples either natural or synthetic, and finally **(III) semi-supervised**, where the training data is a mix of single-source and multi-source samples.

## 4.1 THE UNSUPERVISED SETTING

We have compared our proposed framework on MUSIC dataset with various types of state-of-the-art baselines, as presented in Table 1. For comparison studies on VGGSound and AudioCaps datasets, see Appendix G.4. Here, the performance is measured for different complexities of training scenarios containing $1 \sim 4$ single source components in training mixtures. We use the same test set for all training scenarios containing two components in mixtures. For comparison test results on 3-component mixtures, see Appendix G.5. There are three baseline categories: unconditional, image-conditional, and language-conditional methods. Most conditional models are primarily built on top of the mix-and-separate method introduced in Zhao et al. (2018). We have reproduced the results for all baselines under the same training and test conditions. For a fair comparison, we have incorporated our improved conditional U-Net architecture in most baselines.

**Comparison to Unconditional Methods:** Unconditional methods rely on complex post-processing selection to extract the target sound from the predicted sounds. Following their training methods, we use similar selection methods on test set to find the best match with ground truths which can be a potential limitation in real-time scenarios. PIT (Yu et al., 2017) can only be used in supervised training

on single source sounds that gives comparable results with conditional methods. MixIT (Wisdom et al., 2020) and MixPIT (Karamatlı & Kırbız, 2022) are built for multi-source training on top of the PIT framework. MixIT suffers from over-separation by predicting more components than the actual input mixtures. Though MixPIT reduces the over-separation by directly predicting the mixture in training, it often suffers under-separation on the test set. Our framework, however, achieves $+4.7$ and $+4.3$ SDR improvements compared to MixIt and MixPIT, respectively.

**Comparison to Image Conditional Methods:** The MUSIC dataset also comes with corresponding YouTube videos for audio samples which can be used as visual conditioning signal for audio separation. To this end, we use the mean embedding of multiple sounding source images extracted with corresponding pretrained image encoders for conditioning. CLIPSep (Dong et al., 2022) introduces CLIP-Image encoder for conditioning in the SOP framework (Zhao et al., 2018) substituting its ResNet-18 encoder. CoSep (Gao & Grauman, 2019) further adds a pre-trained object detector for finer conditioning and uses co-separation loss for classifying each sounding source. However, CoSep is limited by the number of training classes and pretraining of the object detectors on the target classes. On 2-source training, our method achieves $+3.8$, $+3.0$, and $+3.3$ SDR improvements compared to SOP, CoSep, and CLIPSep methods, which use our improved U-Net model.

**Comparison to Language Conditional Methods:** CLIPSep (Dong et al., 2022) also includes the CLIP-text encoder which can be used for language conditioning. Additionally, we train separate CLIPSep models with its CLIP-text encoder replaced by frozen Bert (Devlin et al., 2018), and CLAP (Wu et al., 2023) text encoders denoted by BertSep and CLAPSep, respectively. LASS-Net (Liu et al., 2022) is another baseline that uses Bert text encoder with custom ResUnet model architecture. Note that all these methods lack fine-grain conditioning on single source predictions which results in significant performance drop on multi-source training. As for weak supervision, we experimented with Weak-Sup (Pishdadian et al., 2020) which introduces a separate classifier on top of separation models. Apart from restricting the model to certain number of classes, such fine-grain weak supervision often results in spectral loss; plus, without proper pre-training of the classifier on single-source samples, such methods face convergence issues that deteriorates the performance. In comparison, our method consistently achieves significant performance improvements in all challenging multi-source training scenarios over all these baselines. Notably, we achieve $97.5\%$ of the supervised method's performance trained on 2-source mixtures.

**Bi-modal Embedding vs. Timbre Classification:** A key question one might ask is whether we can get similar gain if instead of using the bi-modal embedding CLAP model, we incorporate a simpler timbre classification model to generate weak-supervision for single source prompts, similar to CoSep (Gao & Grauman, 2019) and Weak-Sup (Pishdadian et al., 2020). After all, CLAP can be seen as a zero-shot audio classifier with an open-ended set of classes. To test this idea, we have replaced our CLAP-based loss with a timbre classification-based loss, where the classifier shares the exact same architecture as that of the CLAP audio encoder but the contrastive loss is replaced by the cross-entropy loss. Since, in general, the conditioning prompt can contain more than one classes, we have treated this problem as a *multi-label* classification problem with $C$ binary classification outputs in the last layer, where $C$ is the number of classes. Furthermore, we have trained our classifier-based loss under two scenarios: (I) concurrently with the separation model, and (II) pretrained beforehand. In both cases, the training dataset is the same as that of the original separation task. The results are shown in Table1. As the results show, the concurrent version performs worse than some of the baselines that do not even have weak-supervision. And while the pretrained version does better than the baselines, its performance is still significantly lower than our proposed framework using the CLAP loss, not to mention its restricted applicability due to the fixed number of classes. We hypothesize the superiority of CLAP-based supervision comes from the large-scale pretraining of CLAP which enables us to transfer that knowledge to source separation. In other words, in the limit for large-scale training data and gigantic number of classes, the classification approach should perform as well as the CLAP-based loss, but at that point, we might as well use CLAP.

## 4.2 The Semi-supervised Setting

To demonstrate semi-supervised performance on synthetic mixtures, we form the training set by combining supervised and unsupervised training subsets. The results are given in Table 2. As shown in the table, when we use only $5\%$ of MUSIC and VGGSound datasets as supervised data for both our semi-supervised method and the supervised baseline, while letting our semi-supervised framework use the rest of $95\%$ as unsupervised data, we get the dramatic 6.2 SDR boost over the su-

Table 2: Comparisons of the proposed semi-supervised learning with different portions of single-source and multi-source subsets. **Bold** and blue represents the best and second best performance.

| Training Method | Test Set Mixture | Single-source Data | | Multi-source Mixture Data | | | Performance |
|---|---|---|---|---|---|---|---|
| | | Dataset | Fraction | Dataset | Fraction | #Source | (SDR) |
| Supervised | MUSIC-2Mix | MUSIC | 100% | - | - | - | $8.1 \pm 0.31$ |
| Supervised | MUSIC-2Mix | MUSIC | 5% | - | - | - | $2.6 \pm 0.33$ |
| Unsupervised | MUSIC-2Mix | - | - | MUSIC | 100% | 2 | $7.9 \pm 0.35$ |
| Semi-Supervised | MUSIC-2Mix | MUSIC | 5% | MUSIC | 95% | 2 | $8.8 \pm 0.28$ |
| Semi-Supervised | MUSIC-2Mix | MUSIC | 5% | MUSIC | 95% | 3 | $8.2 \pm 0.22$ |
| Semi-Supervised | MUSIC-2Mix | MUSIC | 5% | MUSIC | 95% | 4 | $7.4 \pm 0.31$ |
| Semi-Supervised | MUSIC-2Mix | MUSIC | 10% | MUSIC | 90% | 2 | $8.9 \pm 0.26$ |
| Semi-Supervised | MUSIC-2Mix | MUSIC | 25% | MUSIC | 75% | 2 | $9.2 \pm 0.24$ |
| Semi-Supervised | MUSIC-2Mix | MUSIC | 75% | MUSIC | 25% | 2 | $9.5 \pm 0.29$ |
| Semi-Supervised | MUSIC-2Mix | MUSIC | 100% | VGGSound | 100% | 2 | $\mathbf{9.9} \pm 0.35$ |
| Semi-Supervised | MUSIC-2Mix | VGGSound | 100% | MUSIC | 100% | 2 | $9.7 \pm 0.35$ |
| Semi-Supervised | MUSIC-2Mix | VGGSound | 100% | MUSIC | 100% | 3 | $9.2 \pm 0.31$ |
| Semi-Supervised | MUSIC-2Mix | VGGSound | 100% | MUSIC | 100% | 4 | $8.9 \pm 0.42$ |
| Supervised | VGGSound-2Mix | VGGSound | 100% | - | - | - | $2.3 \pm 0.23$ |
| Supervised | VGGSound-2Mix | VGGSound | 5% | - | - | - | $0.4 \pm 0.35$ |
| Unsupervised | VGGSound-2Mix | - | - | VGGSound | 100% | 2 | $2.2 \pm 0.29$ |
| Semi-Supervised | VGGSound-2Mix | VGGSound | 5% | VGGSound | 95% | 2 | $3.1 \pm 0.31$ |
| Semi-Supervised | VGGSound-2Mix | VGGSound | 75% | VGGSound | 25% | 2 | $\mathbf{3.4} \pm 0.26$ |
| Unsupervised | AudioCaps-2Mix | - | - | AudioCaps | 100% | 1∼6 | $2.9 \pm 0.23$ |
| Semi-Supervised | AudioCaps-2Mix | VGGSound | 100% | AudioCaps | 100% | 1∼6 | $\mathbf{4.3} \pm 0.34$ |

pervised method. This shows that our semi-supervised framework can make a significant difference in scenarios where well-curated, single source samples are scarce and costly to acquire by leveraging a large corpora of unsupervised data. More interestingly, even when we let the supervised baseline use $100\%$ of the data as supervised single-source data, our semi-supervised approach still beats it by $+0.7$, and $+0.8$ SDR boost on the two datasets using only $5\%$ of the supervised data! Based on this result, we hypothesize that, in addition to data augmentation for training, our proposed framework offers a powerful regularization mechanism that boosts the generalization of the supervised method by encouraging it to also discover salient patterns from unsupervised data. For a more comprehensive study of our framework's regularization effects, see Appendices G.2, G.5, and G.7.

Finally, we note that for realistic, natural mixture datasets where single source audio samples are not available for semi-supervised learning, we can still utilize our semi-supervised scheme by running it across multiple datasets and let supervised and unsupervised training samples come from different datasets. To this end, we have trained our model using the AudioCaps natural mixtures as the unsupervised subset and the VGGSound dataset as the supervised subset, and have achieved $+1.4$ SDR boost over the weakly-supervised baseline trained on AudioCaps only.

## 5 CONCLUSION AND FUTURE WORK

In this paper, we proposed a weakly supervised learning framework for language conditional audio separation when single source audio samples are not available during training. By leveraging cross modal similarity between the audio and language modalities through the pretrained CLAP model, our methodology is capable of generating weak supervision signal for single-source prompts during training, which in turn, enables the model to reduce the shift between the training and test data distributions. By conducting extensive experiments, we demonstrate the superiority of our proposed framework over the SOTA baselines by significant margin, shrinking the gap between unsupervised and supervised methods. Furthermore, by incorporating our framework into the semi-supervised setting, we showed that our framework beats the supervised method itself. More interestingly, as shown by the experiments, our framework still maintains its superiority even when the supervised baseline had access to 20x more supervised single source data than our framework. This latter result highlights the ability of our framework to employ natural regularization through the language modality during training.

As mentioned earlier, aside from the implementation details, the core idea of our proposed framework is generic and modality-independent, and therefore can be adopted for conditional segmentation tasks in other modalities as well (*e.g.*, unsupervised image segmentation in Computer Vision). This property provides a fertile ground for future research directions and novel applications.

ACKNOWLEDGEMENTS

This work was supported in part by ONR Minerva program, iMAGiNE - the Intelligent Machine Engineering Consortium at UT Austin, and a UT Cockrell School of Engineering Doctoral Fellowship.

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

## A OVERVIEW OF SUPPLEMENTARY MATERIALS

We cover the following topics in our supplementary materials:

- All notations used in the paper are summarized in Appendix B.
- The proposed conditional U-Net architecture is detailed in Appendix C.
- The details of the training process for the proposed framework are illustrated in Appendix D.
- The datasets' details and the data prepossessing pipeline are presented in Appendix E.
- The details of evaluation metrics are presented in Appendix F.
- Additional experimental ablation studies are reported in Appendix G.
- And finally, a qualitative analysis of the model's performance can be found in Appendix I.

Table 3: The glossary of all notations used in the paper.

| Type | Description | Notation |
|---|---|---|
| Scalar Parameters | Total number of mixtures in dataset | $P$ |
| | Number of sampled mixtures in a batch | $N$ |
| | Number of single source components in a mixture | $K$ |
| | Temperature parameter in contrastive loss | $\tau$ |
| | Unsupervised reconstruction loss | $\mathcal{L}_{URL}$ |
| | Contrastive loss | $\mathcal{L}_{CNT}$ |
| | Consistency reconstruction loss | $\mathcal{L}_{CRL}$ |
| | Total weak-supervision loss | $\mathcal{L}_{TWL}$ |
| | Total semi-supervised learning loss | $\mathcal{L}_{SSL}$ |
| | Weak supervision loss weights | $\alpha, \beta, \gamma$ |
| | Semi-supervised learning loss weights | $\lambda_s, \lambda_u$ |
| Vectors/Matrices | Complete dataset | $\mathcal{D}$ |
| | Batch of mixtures | $\mathcal{B}$ |
| | Batch of synthetic mixture-of-mixturess | $\mathcal{B}'$ |
| | $i^{th}$ Sound mixture | $\mathcal{M}_i$ |
| | Mixture of mixture | $\mathcal{M}'$ |
| | Language prompt of $i^{th}$ mixture | $\mathcal{T}_i$ |
| | The $k^{th}$ single source component in the $i^{th}$ mixture | $\mathcal{S}_i^k$ |
| | Language prompts of single source sound | $\mathcal{T}_i^k$ |
| Models | Frozen CLAP language encoder | $\varepsilon_L(\cdot)$ |
| | Frozen CLAP audio encoder | $\varepsilon_A(\cdot)$ |
| | Conditional U-Net audio source separation model | $f_\theta(\cdot)$ |
| | Conditional U-Net mask model | $g_\theta(\cdot)$ |
| Operators/Functions | Magnitude function | $|\cdot|$ |
| | Phase function | $\phi(\cdot)$ |
| | Short Term Fourier Transform | $S(\cdot)$ |
| | Softmax function with temperature parameter $\tau$ | $\zeta_\tau(\cdot)$ |
| | Audio-language Cosine Similarity | $c_{ikjt}$ |
| | L1 loss | $\|\cdot\|_{\ell_1}$ |
| | Hadamard product | $\odot$ |

## B THE NOTATION GLOSSARY

Table 3 presents the glossary of all notations used in the paper. We have divided the notations into four groups: scalars, vectors/matrices, models, and operators/functions.

## C THE PROPOSED ARCHITECTURE

### C.1 THE LANGUAGE-CONDITIONAL U-NET

To extract rich features for faithful reconstruction of the audio sources conditioned on the input prompt, we propose an enhanced conditional U-Net architecture. Our U-Net model operates on the magnitude spectrum of input mixtures, and estimates the segmentation mask for the corresponding

source(s) based on conditional feature embedding. Prior works on conditional sound separation, mostly used unconditional U-Net with post conditioning on final generated features from the U-Net (Dong et al., 2022; Zhao et al., 2018). Some works (Gao & Grauman, 2019) used simple conditional feature concatenation at the innermost layer of the U-Net. Since most of these methods are primarily built for supervised separation, which is a much simpler problem, the vanilla U-Net architecture is often sufficient. However, since post-conditioning methods cannot leverage the conditional language features through the network, their performance can degrade significantly in the unsupervised setting. To overcome this, we redesign the conditional U-Net architecture by introducing multi-scale cross attention conditioning on the intermediate feature maps of the U-Net. The architecture is shown in Figure 5.

We incorporate three main building blocks into the proposed conditional U-Net: residual block (ResBlock), self-attention (SA), and cross-attention (CA) modules. The residual block is used for enhancing model capacity following He et al. (2016) at every scale of feature processing. For the input $x$, the operation can be represented by,

$$x = x + \text{ConvBlock}(x) \tag{8}$$

where *ConvBlock* represents two successive convolutional layers. The self-attention and cross-attention modules are designed using the multi-head attention (MHA) operations introduced by (Vaswani et al., 2017). Self-attention re-calibrates the feature space before applying the conditioning modulation. The self-attention operation for input $x$ is given by,

$$x = x + \text{MHA}(Q = x, K = x, V = x) \tag{9}$$

where $Q, K, V$ represent query, key, and values used in the *MHA* operation, respectively. In contrast, cross-attention selectively filters the relevant features based on conditional features. For condition embedding $y$ with input $x$, the cross-attention mechanism is given by,

$$x = x + \text{MHA}(Q = x, K = y, V = y) \tag{10}$$

We divide the conditional U-Net model into two sub-networks: the *Head* network and the *Modulator* network. The Head network operates on the fine-grain features of the higher signal resolutions to generate coarse-grain features to be conditioned later by the language modality. Only ResBlocks with traditional skip connections are used at each scale of the Head network. In contrast, the Modulator network applies the feature modulation based on the conditional language embedding. We incorporate the self-attention and cross-attention operations in the skip connections of every Modulator network layer. In total, the U-Net contains 7 layers of encoder and decoder. The Head network contains top four layers of encoding and decoding, and the Modulator network contains the remaining three layers. Table 4 shows the architectural details of each block in our enhanced conditional U-Net.

## C.2 THE INFERENCE PIPELINE

For the inference, we use the similar pipeline as baseline methods. As shown in Figure 4, the conditional U-Net takes the input mixture and the language prompt (querying for the target source), and generates the (soft) magnitude mask. The mask is applied on the mixture's magnitude spectrogram, while the phase is directly copied from the input.

## D THE TRAINING DETAILS

All the models are trained for 50 epochs with initial learning rate of 0.001. The learning rate drops by the factor of 0.1 after every 15 epochs. Adam optimizer (Kingma & Ba, 2014) is used with $\beta_1 = 0.9$, $\beta_2 = 0.999$ and $\epsilon = 10^{-8}$ for backpropagation. All the training was carried out with 8 RTX-A6000 GPUs with 48GB memory. We validate the model after every training epoch. We use the batch size of 32 for the MUSIC dataset, and batch size of 64 for the VGGSound and AudioCaps datasets. We reproduce all baselines under the same settings. PyTorch library (Paszke et al., 2019) is used to implement all the models. The complete training algorithm of the proposed framework is illustrated in Algorithm 1.

Furthermore, in Figure 6, we have visualized the detailed loss curves over the training course of the proposed weakly supervised training (Fig. 6a) and its semi-supervised flavor (Fig. 6b). We have

Table 4: Architectural details of proposed building blocks in the conditional U-Net model.

| Block Type | Parameters | Values |
|---|---|---|
| Cross Attention Block (CA) | No. of attention heads | 8 |
| | No. of channels | 512 |
| | Head dimension | 64 |
| | Condition dimension | 77x768 |
| | No. of linear layers in attention | 4 |
| | Attention type | Softmax |
| | Normalization Layer | LayerNorm |
| | No. of linear layers in MLP | 2 |
| | MLP intermediate channels | 1024 |
| | MLP intermediate activation | GeLU |
| Self Attention Block (SA) | Num of attention heads | 8 |
| | Channel dimension | 512 |
| | Head dimension | 64 |
| | No. of linear layers | 4 |
| | Attention type | Softmax |
| | Normalization layer | LayerNorm |
| | No. of linear layers in MLP | 2 |
| | MLP intermediate channels | 1024 |
| | MLP intermediate activation | GeLU |
| Residual Block (ResBlock) | Conv kernel size | (3, 3) |
| | No. of convolutions | 1 |
| | Normalization layer | BatchNorm |
| | Activation | Leaky ReLU (th=0.2) |
| | Channel expansion ratio | 1 |
| Encoder Downsampler Module | Operator | Strided Convolution |
| | Kernel size | 4x4 |
| | Strides | 2x2 |
| | Channel expansion ratio | 2 |
| Decoder Upsampler Module | Spatial upsampler | Bilinear upsampling |
| | Scale | 2.0 |
| | Channel compressor | Convolution |
| | Channel compression ratio | 2.0 |

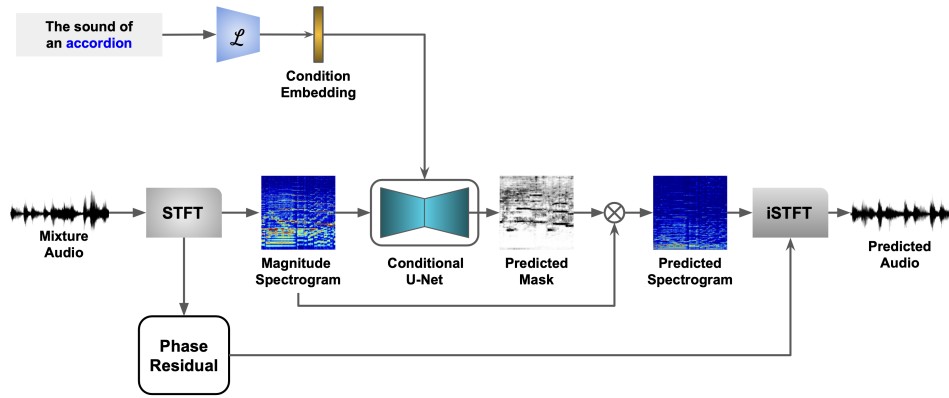

Figure 4: Inference pipeline for the proposed language conditional sound separation framework.

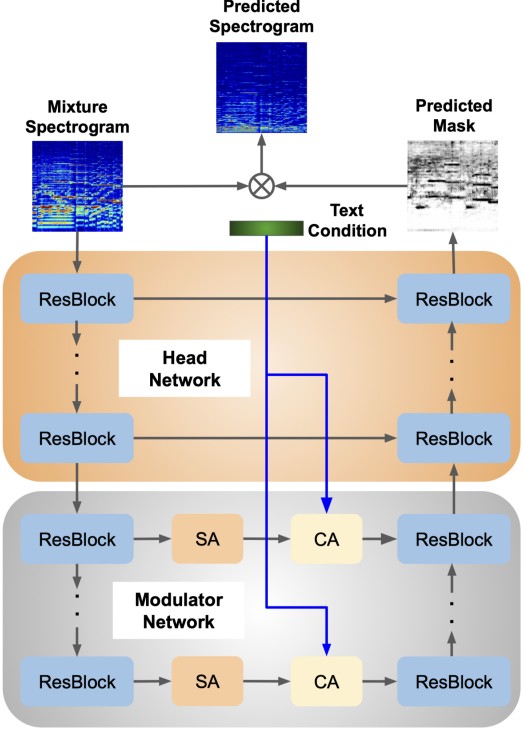

Figure 5: Proposed conditional U-Net architecture. We incorporate three building blocks: residual block (ResBlock), self-attention (SA), and cross-attention (CA) blocks. The model is divided into two modules: the head and the modulator. The *head* network operates on fine-grained features and generates latent embedding. The *modulator* network modulates latent features based on cross-attention conditioning.

combined both unsupervised reconstruction loss $\mathcal{L}_{URL}$ and consistency reconstruction loss ($\mathcal{L}_{CRL}$) in the reconstruction loss plot. For further analysis of the loss components as well as their weight hyper-parameter tuning details, please refer to Appendix G.2.

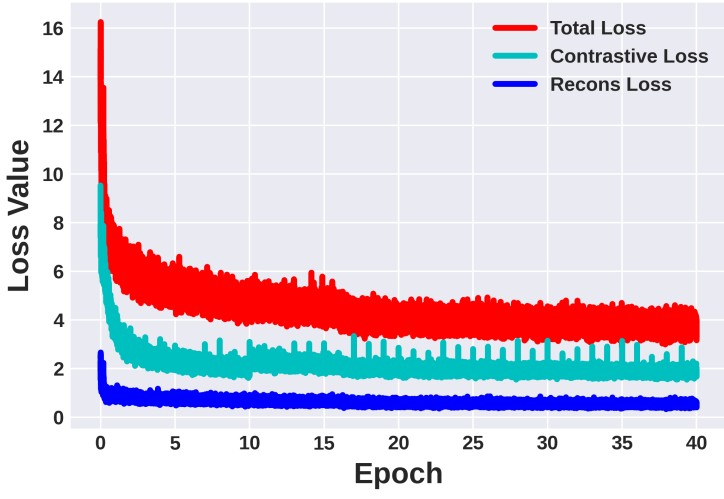

(a) Weakly supervised training

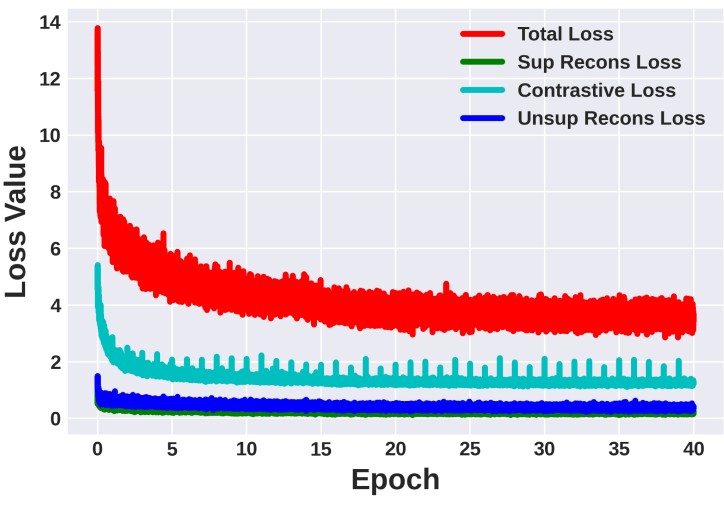

(b) Semi-supervised training

Figure 6: Visualization of loss curves over training epochs in proposed (a) weakly supervised training, and (b) semi-supervised training.

---

**Algorithm 1** The Proposed Weakly Supervised Training for Audio Source Separation

---

**Input:** Dataset $\mathcal{D} = \{\mathcal{M}_i, \mathcal{T}_i\}_{i=1}^P$, Single source text prompts $\{\mathcal{T}_i^k\}_{k=1}^K$, $K$: # of sources per mixture, masking U-Net $g_\theta$, Pre-trained joined embedding encoders $(\varepsilon_L, \varepsilon_A)$.
**Require:** Initialize weights of $g_\theta$, keep pre-trained joint embedding encoders $(\varepsilon_L, \varepsilon_A)$ frozen.
 1: **for** $t \in [1, T]$ **do**                                                          ▷ T: Training iteration
 2:     Sample $N$ Mixture with text prompts $\{\mathcal{M}_n, \mathcal{T}_n\}_{n=1}^N \in \mathcal{D}$                       ▷ Batch size $\leftarrow N$
 3:     **for** $n \in [1, N]$ **do**
 4:         Sample another mixture $\{\mathcal{M}_m, \mathcal{T}_m\}$   ▷ No single sound source overlaps in $\mathcal{M}_n$ , $\mathcal{M}_m$
 5:         Prepare MoM $\mathcal{M}' \leftarrow \mathcal{M}_n + \mathcal{M}_m$
 6:         Predict $\widehat{\mathcal{M}}_n \leftarrow f_\theta(\mathcal{M}', \mathcal{T}_n)$
 7:         Predict $\widehat{\mathcal{M}}_m \leftarrow f_\theta(\mathcal{M}', \mathcal{T}_m)$
 8:         Compute $\mathcal{L}_{URL}(\mathcal{M}; \theta)$ with $(\mathcal{M}_n, \widehat{\mathcal{M}}_n; \mathcal{M}_m, \widehat{\mathcal{M}}_n)$ using  equation 2
 9:         **for** $k \in [1, K]$ **do**
10:             Compute single source sound $\widehat{\mathcal{S}}_n^k \leftarrow f_\theta(\mathcal{M}_n, \mathcal{T}_n^k)$
11:             Compute single source sound $\widehat{\mathcal{S}}_m^k \leftarrow f_\theta(\mathcal{M}_m, \mathcal{T}_m^k)$
12:         **end for**
13:         Compute $\mathcal{L}_{CNT}(\mathcal{M}', \theta)$ with $\{\widehat{\mathcal{S}}_n^k, \mathcal{T}_n^k; \widehat{\mathcal{S}}_m^k, \mathcal{T}_m^k\}_{k=1}^K$ using  equation 3 and equation 4
14:         Reconstruct mixture $\widetilde{\mathcal{M}}_n \leftarrow \sum_{k=1}^K \widehat{\mathcal{S}}_n^k$ and $\widetilde{\mathcal{M}}_m \leftarrow \sum_{k=1}^K \widehat{\mathcal{S}}_m^k$
15:         Compute $\mathcal{L}_{CRL}(\mathcal{M}', \theta)$ with $(M_n, \widetilde{M}_n; \mathcal{M}_m, \widetilde{\mathcal{M}}_m)$ using equation 5
16:     **end for**
17:     Compute total loss $\mathcal{L}_{TWL}$ using $\mathcal{L}_{URL}, \mathcal{L}_{CRL}$, and $\mathcal{L}_{CNT}$ for all $N$ mixtures
18:     Back-propagate $\nabla\mathcal{L}_{TWL}$ and update weights of $g_\theta$
19: **end for**

---

# E   DATASET PREPARATION

## E.1   DATASETS DESCRIPTION

**MUSIC Dataset**   Following prior works, we experiment with MUSIC dataset (Zhao et al., 2018) for musical instrument separation task. Instead of using the original 11 instrument datasets, we use its extended version of MUSIC-21 containing around $1,200$ videos from 21 musical instruments. Since some videos are not available, our aggregated version contains $1,086$ videos in total. The video duration ranges $1 \sim 5$ minutes. We extract audios and class labels annotations from each video. We use $80\%$ videos of each classes for training and the remaining for testing. For training, we randomly sample around 6s duration segments from each audio, while for testing, we prepare non-overlapping samples from the whole length audio. The dataset only contains sounds of single-source musical instruments. To use this datset for unsupervised training, we create synthetic training mixtures by sampling different combinations of $K$ single source sounds. Text prompts are then generated using the class labels of the single source sounds. We use the common template for representing the single and multi-source language condition prompts, as presented in Table 6.

**VGGSound Dataset**   VGGSound (Chen et al., 2020) is a large-scale environmental sound datasets containing more than $190,000$ videos from 309 classes. Since many corresponding videos are not available in YouTube, our aggregated subset contains $175,599$ videos. We use the official train and test split of VGGSound that contains $162,199$ training videos and $13,398$ test videos. Every video contains an audio, mostly with a single source. Each video duration is 10s that contains single source audio collected from different environments corrupted by natural noise. The sounding event duration varies from $1 \sim 10$s. Because of that, we use the full-length audio samples in VGGSound for our experiments. In order to use VGGSound for the unsupervised learning scenario, we mix $K$ random single source samples for each training mixture. The corresponding text prompts are generated using the class labels of the the sounding sources in the mixture, similar to Table 6.

**AudioCaps Dataset**   AudioCaps (Kim et al., 2019) contains around $50,000$ *natural* sound mixtures of 10s duration each. It also includes the complete captions of the prominent sources in each mixture. We use the official train and test splits that contain $45,182$ and $4,110$ mixtures, respec-

tively. In general, each mixture contains $1 \sim 6$ single source components. To cover all sounding events included in the text caption, we use full-length audios of 10s. We use Constituent-Tree library (Halvani, 2023) to extract fine-grain phrases representing each sounding source from the full caption. We initially extract several sentence and noun phrases, then perform simple post-processing on them to eliminate the overlapping phrases. Some examples of extracted phrases from the full captions are given in Table 5. To handle different number of mixture components, we sample a fixed number of phrases from each caption. In case there are not enough sounding phrases in the text prompt, we re-sample some of the phrases, and introduce weighted reconstruction to ensure proper reconstruction of the mixture.

AudioCaps is primarily used to measure training performance on natural mixtures containing diverse sounding events, as opposed to synthetic mixtures. However, to evaluate the performance of the model, we prepare synthetic mixture-of-mixtures by combining two mixtures from the AudioCaps test set. At the test time, the model is queried with one full-length caption representing one of the mixtures in synthetic MoM, and evaluated using the corresponding mixture.

### E.2 DATA PREPROCESSING PIPELINE

We use the sampling rate of 11kHz for audio samples in all datasets. Only mono-channel audio is used. The audio clip length is chosen to be $65,535$ for MUSIC dataset, and $110,000$ for the AudioCaps and VGGSound datasets. Since AudioCaps and VGGSound datasets are noisy, and usually contain sounding regions on small portion of 10s duration, we use full length audio samples for these two datasets. For the MUSIC dataset, we extract consecutive $65,535$ segments from the complete duration of the samples representing 6s audio. We compute the spectrogram for each sample using short-term Fourier transform (STFT) with a window size of $1024$, a filter length of $1024$, and a hop size of $256$.

The CLAP model is pre-trained with 10s duration audios of 48KHz sampling rate and has different pre-processing pipeline than ours. To integrate the pretrained CLAP model in our training pipeline, we initially reconstruct the sound waveform from the predicted spectrogram. For audio samples extracted from MUSIC dataset that contains 6s duration segments, we repeat the waveform to extract equivalent 10s duration of audios. Then, the audio waveform is resampled with 48KHz sampling rate. We use Torchaudio package (Yang et al., 2022) to process predicted audio samples in the training loop. To estimate the contrastive loss ($\mathcal{L}_{CNT}$) with the CLAP model, we follow the same pipeline of CLAP with the pretrained temperature value for $\tau$. We note that the CLAP model is kept frozen throughout the entire training, as it is only used to generate weak supervision. For text conditioning signals, instead of the projected mean-pooled token representation of language prompts, we use the complete language embedding of dimension ($77 \times 768$) representing 77 tokens, generated by the CLAP language encoder.

## F EVALUATION METRICS

We use three evaluation metrics in our experiments: SDR, SIR, and SAR. Here, we provide the detailed equations as well as the explanation of each metric. We note that a predicted sound $\mathcal{S}_{pred}$ can be represented as a combination of the true sound $\mathcal{S}_{true}$, the interference of other sources in the mixture $\mathcal{E}_{interf}$, and the artifacts generated during reconstruction $\mathcal{E}_{artifact}$; that is, $\mathcal{S}_{pred} = \mathcal{S}_{true} + \mathcal{E}_{interf} + \mathcal{E}_{noise} + \mathcal{E}_{artifact}$. According to Vincent et al. (2006), these evaluation metrics are described as follows:

**Signal-to-Distortion Ratio (SDR):** SDR is the primary metric used for evaluating sound separation performance in most prior work. It represents the overall measure of the sound quality considering all kinds of distortions. It is given by

$$\text{SDR} = 10 \log_{10} \frac{\|\mathcal{S}_{true}\|^2}{\|\mathcal{E}_{interf} + \mathcal{E}_{noise} + \mathcal{E}_{artifact}\|^2} \tag{11}$$

**Signal-to-Interference Ratio (SIR):** SIR is also widely used evaluation metric in sound separation. It represents the "leakage" or "bleed" from other sounding sources in the mixture to the

Table 5: Examples of some extracted phrases from full-length AudioCaps (Kim et al., 2019) captions.

| Complete Audio Captions | Extracted Phrases |
|---|---|
| A young female speaks, followed by spraying and a female screaming | A young female speaks. Spraying and a female screaming |
| Motor noise is followed by a horn honking and a siren wailing | Motor noise. A horn honking. A siren wailing. |
| Rustling occurs, ducks quack and water splashes, followed by an adult female and adult male speaking and duck calls being blown | Rustling occurs. Ducks quack and water splashes. An adult female and adult male speaking. Duck calls being blown. |
| An audience gives applause as a man yells and a group sings | An audience gives applause. A man yells. A group sings. |
| A man speaks over intermittent keyboard taps | A man speaks. Intermittent keyboard taps. |
| An airplane engine runs | An airplane engine. |

predicted sound. SIR measures the quality of the predicted sound considering the amount of cross-interference from other sources. It is given by

$$\text{SIR} = 10 \log_{10} \frac{\|\mathcal{S}_{true}\|^2}{\|\mathcal{E}_{interf}\|^2} \tag{12}$$

**Signal-to-Artifact Ratio (SAR):** SAR is mostly used to measure how realistic the predicted sound is. It measures the amount of synthetic artifacts present in the predicted audio. Without any separation applied, the original mixture usually have very high SAR, as it does not contain that many of artifacts. However, as the model learns to separate single source components from the mixture, it is expected to introduce more artifacts.

$$\text{SAR} = 10 \log_{10} \frac{\|\mathcal{S}_{true} + \mathcal{E}_{interf} + \mathcal{E}_{noise}\|^2}{\|\mathcal{E}_{artifact}\|^2} \tag{13}$$

In order to calculate these metrics, we have used the Python package **torch-mir-eval** (Montesinos, 2021) which is the Pytorch implementation of **mir-eval** (Raffel et al., 2014).

## G  ADDITIONAL EXPERIMENTAL STUDIES

In this section, we present additional experimental ablation studies for a deeper analysis of the proposed framework compared to the state-of-the-art baseline methods as well as some of our design choices.

### G.1  ABLATION STUDY ON CONDITIONAL U-NET ARCHITECTURE

We have studied the contribution of all three building blocks in the proposed conditional U-Net architecture. We have experimented with both supervised and unsupervised training settings on

Table 6: Text query templates with examples for single and synthetic multi-source sounds

| Source | Query Template | Example |
|---|---|---|
| Single Source | The sound of {source} | The sound of **guitar**. |
| Multi-Source (2-Source) | The sound mixture of {source-1} and {source-2} | The sound mixture of **guitar** and **piano**. |
| Multi-Source (3-Source) | The sound mixture of {source-1}, {source-2}, and {source-3} | The sound mixture of **guitar**, **piano**, and **violin**. |

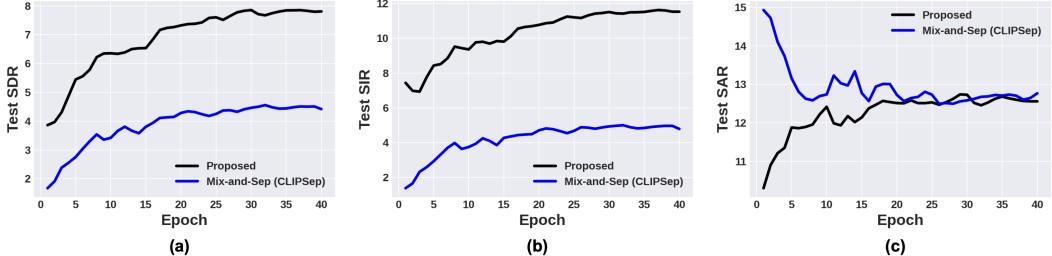

Figure 7: Test metric plots during training with two-source mixtures on MUSIC dataset. We use CLIPSep as the mix-and-separate baseline method. Our proposed method achieves significant improvement in terms of SDR and SIR by largely reducing noise and cross-interference in predictions, respectively. As the model tries to learn to separate single-source sounds, some artifacts are introduced, which in turn result in low SAR value in the early stages of training. However, most of these artifacts get removed over the course of training which in turn causes SAR to increase to the same level as the baseline method by the end of training. In other words, by the end of training, both our method and the baseline produce audio samples with reasonable quality regardless of their separation performance.

Table 7: Ablation on three building blocks of proposed conditional U-Net architecture: ResBlock, self-attention (SA), and cross-attention(CA). The vanilla U-Net contains single convolutional layer instead of ResBlock, and simple skip connections instead of attention modules. Test SDR on 2-source mixture is reported for various single and multi-source training scenarios on MUSIC dataset. For the single source training, simple *mix-and-separate* based on CLAPSep is used. For the multi-source training, proposed weakly supervised training is used. All three blocks contribute to considerable performance gain mostly in challenging multi-source scenarios. **Bold** and blue represents the best and second best performance in each group, respectively.

| ResBlock | SA | CA | Total Params (M) | Single Source | 2-Source | 3-Source | 4-Source |
|---|---|---|---|---|---|---|---|
| ✗ | ✗ | ✗ | 30.7M | 7.4 | 6.7 | 5.8 | 4.9 |
| ✗ | ✗ | ✓ | 37.4M | 7.7 | 7.1 | 6.3 | 5.5 |
| ✗ | ✗ | ✓✓ | 44.7M | 7.8 | 7.3 | 6.5 | 5.7 |
| ✗ | ✓ | ✗ | 36.6M | 7.6 | 6.9 | 6.1 | 5.3 |
| ✗ | ✓✓ | ✗ | 42.9M | 7.5 | 6.8 | 5.9 | 5.1 |
| ✓ | ✗ | ✗ | 74.3M | 7.6 | 6.9 | 6.1 | 5.2 |
| ✓✓ | ✗ | ✗ | 118.8M | 7.1 | 6.4 | 5.6 | 5.0 |
| ✗ | ✓ | ✓ | 43.7M | 7.9 | 7.4 | 6.7 | 5.8 |
| ✗ | ✓✓ | ✓✓ | 57.7M | 7.9 | 7.2 | 6.6 | 5.9 |
| ✓ | ✓ | ✓ | 81.4M | **8.1** | **7.9** | **7.1** | **6.2** |

the MUSIC dataset. The test set contains 2-source mixtures as before. The baseline vanilla U-Net contains single convolutional layer instead of ResBlock, and direct skip connections instead of self-attention and cross-attention modules. The results are given in Table 7. Utilizing all three building blocks results in +0.7, +1.2, +1.3, and +1.3 SDR improvements on single source, 2-source, 3-source, and 4-source training settings, respectively. We note that the performance improvements are comparatively higher in the challenging unsupervised setting compared to the supervised setting. Since unsupervised training is mostly guided by weak supervision generated by the bi-modal CLAP model, we hypothesize that the multi-scale feature modulation based on conditional embedding becomes more critical in such cases.

Note that the increased number of parameters by the way of adding new blocks can be seen as the major contributor to the performance gain. To control for this effect, in some of the ablation scenarios, we have inserted the target block(s) twice in a row merely to increase the capacity of the model (shown by ✓✓ in Table 7). As the results show, while increasing the number of model's parameters contributes to the performance gain, it is *not* the major driver. In fact, some of our smaller candidates beat the larger ones by simply incorporating a new block type. This shows that

Table 8: Ablation on loss components of proposed weakly supervised training method with multi-source training mixtures from the MUSIC dataset. Test SDR on 2-source mixtures is reported for all cases. Unsupervised reconstruction loss ($\mathcal{L}_{URL}$) underperforms in higher mixtures due to the lack of fine-grain supervision. Contrastive loss ($\mathcal{L}_{CNT}$), on the other hand, produces weak supervision that performs the best combined with proposed consistency reconstruction loss ($\mathcal{L}_{CRL}$). Combining all three loss components achieves significant performance improvements. **Bold** and blue represents the best and second best performance in each group, respectively.

| $\mathcal{L}_{\mathbf{URL}}$ | $\mathcal{L}_{\mathbf{CNT}}$ | $\mathcal{L}_{\mathbf{CRL}}$ | 2-Source | 3-Source | 4-Source |
|---|---|---|---|---|---|
| ✓ | ✗ | ✗ | 5.5 | 4.3 | 3.5 |
| ✗ | ✓ | ✗ | 4.9 | 2.8 | 1.7 |
| ✗ | ✗ | ✓ | 2.4 | 1.3 | 0.6 |
| ✗ | ✓ | ✓ | 6.8 | 5.9 | 5.3 |
| ✓ | ✓ | ✓ | **7.9** | **7.1** | **6.2** |

Table 9: Ablation study on loss weights with proposed weakly supervised and semi-supervised learning. For the weakly supervised training, 2-source mixtures from the (a) MUSIC dataset, (b) VGGSound dataset, and (c) natural mixtures from the AudioCaps dataset are used. (d) For the semi-supervised training, 5% single-source and 95% 2-source mixtures are used. Test SDR on 2-source mixture separation is reported. **Bold** and blue represents the best and second best performance in each group, respectively.

(a) Weakly supervised training on MUSIC

| $\alpha(\mathcal{L}_{\mathbf{CNT}})$ | $\beta(\mathcal{L}_{\mathbf{CRL}})$ | $\gamma(\mathcal{L}_{\mathbf{URL}})$ | **SDR** |
|---|---|---|---|
| 0.1 | 5 | 5 | 7.7 |
| 0.1 | 5 | 10 | **7.9** |
| 0.1 | 5 | 15 | 7.7 |
| 0.2 | 5 | 10 | 7.8 |
| 0.1 | 10 | 10 | 7.6 |

(b) Weakly supervised training on VGGSound

| $\alpha(\mathcal{L}_{\mathbf{CNT}})$ | $\beta(\mathcal{L}_{\mathbf{CRL}})$ | $\gamma(\mathcal{L}_{\mathbf{URL}})$ | **SDR** |
|---|---|---|---|
| 0.2 | 0.5 | 10 | 2.0 |
| 0.2 | 1 | 10 | 2.1 |
| 0.2 | 2 | 10 | **2.2** |
| 0.1 | 5 | 15 | 1.9 |
| 0.3 | 5 | 5 | 1.7 |

(c) Weakly supervised training on AudioCaps

| $\alpha(\mathcal{L}_{\mathbf{CNT}})$ | $\beta(\mathcal{L}_{\mathbf{CRL}})$ | $\gamma(\mathcal{L}_{\mathbf{URL}})$ | **SDR** |
|---|---|---|---|
| 0.2 | 0.5 | 10 | 2.8 |
| 0.2 | 1 | 10 | **2.9** |
| 0.2 | 2 | 10 | 2.7 |
| 0.1 | 5 | 15 | 2.5 |
| 0.3 | 5 | 5 | 2.4 |

(d) Semi-supervised training on MUSIC

| $\lambda_s(\mathcal{L}_{\mathbf{URL}})$ | $\lambda_u(\mathcal{L}_{\mathbf{TWL}})$ | **SDR** |
|---|---|---|
| 5 | 0.5 | 8.5 |
| 5 | 1 | **8.8** |
| 5 | 2 | 8.6 |
| 2.5 | 1 | 8.7 |
| 0.5 | 1 | 8.2 |

our proposed blocks encode important inductive bias for our problem which can boost the model performance without overfitting.

Furthermore, for a fair comparison with the baseline methods, we have reproduced most of the prior work with our improved U-Net architecture, as shown in Table 1 and 12. We observe consistent improvements of performance by leveraging our modified U-Net. Nonetheless, the improved U-Net architecture increases computational burden in general, but the proposed weakly supervised training can still be applied in resource constrained scenarios by simply using the vanilla U-Net architecture.

## G.2 EFFECTS OF DIFFERENT LOSS COMPONENTS

We have also studied the effects of different loss components in the proposed framework under the challenging unsupervised settings, as shown in Table 8. By only using the unsupervised reconstruction loss ($\mathcal{L}_{URL}$), we get sub-optimal performance due to the training and test distribution shift. On the other hand, the contrastive loss by itself ($\mathcal{L}_{CNT}$) results in performance drops with significant spectral loss due to the lack of fine-grain supervision to reconstruct the target signal. Similarly, the consistency reconstruction loss($\mathcal{L}_{CRL}$) by itself suffers from convergence issues due to the lack of any supervision to encourage the model for conditional single source separation. In other words,

since the final reconstruction administered by the consistency reconstruction loss ($\mathcal{L}_{CRL}$) greatly depends on the quality of single-source predictions, a significant performance drop is inevitable without using any single-source supervision. However, by combining $\mathcal{L}_{CNT}$ and $\mathcal{L}_{CRL}$, we achieve +1.3, +1.6, and +1.8 SDR improvements over the $\mathcal{L}_{URL}$-only scenario for 2-source, 3-source, and 4-source training settings, respectively. Furthermore, by combining all three losses, we achieve significant performance improvements of +2.4, +2.8, +2.7 SDR over the $\mathcal{L}_{URL}$-only approach for 2-source, 3-source, and 4-source training settings, respectively.

In addition to the elimination study of different loss terms, we have performed an ablation study on 2-component mixtures from the MUSIC and VGGSound datasets, as well as on natural mixtures of AudioCaps dataset, to find the optimal relative weights of these components (i.e. $\alpha$, $\beta$, and $\gamma$ in equation 6). Table 9(a), 9(b), and 9(c) shows these results. It is interesting to observe that, in the optimal setting, $\mathcal{L}_{CNT}$ is weighed two orders of magnitude less than $\mathcal{L}_{URL}$, which suggests that the weak-supervision mechanism in our framework acts as an effective regularizer while the backprop-agated supervision signal is mostly dominated by the reconstruction error. And yet this relatively small regularization effect makes a significant improvement to the final performance of the model during inference. Moreover, VGGSound, and AudioCaps dataset contain significant amount of environmental noises that result in noisy training particularly with the consistency reconstruction losses. As a result, the corresponding weight $\gamma$ of the consistency reconstruction loss $\mathcal{L}_{CRL}$ is relatively reduced in VGGSound and AudioCaps dataset, while $\mathcal{L}_{CNT}$ coefficient $\alpha$ is slightly increased for the best performance. Similar study has been performed to find the optimal relative weights of the supervised and weakly-supervised components for the semi-supervised loss ($\mathcal{L}_{SSL}$) (i.e. $\lambda_s$ and $\lambda_u$ in equation 7). The results are summarized in Table 9(b).

### G.3 Analysis of evaluation metrics for unsupervised training

The metric plots in Figure 7 demonstrate comparative analysis of the evaluation metric curves during the unsupervised (2-source) training. To represent the baseline *mix-and-separate* framework, we have used the CLIPSep (Dong et al., 2022) method. The mix-and-separate baseline attempts to extract the single-source components from the mixture without having any supervision on single-source predictions during training; this results in large noise and cross-interference. In contrast, our proposed weakly-supervised method significantly reduces noise and cross-interference during unsupervised training by leveraging weak supervision through the language modality, and results in a much higher SDR (Figure 7(a) and SIR(Figure 7(b), respectively. However, separating single-source components is susceptible to producing artifacts that cause lower SAR in the early stages of training, as shown in Figure 7(c). Nevertheless, as the training continues, such artifacts are largely eliminated which subsequently improves SAR. In general, SAR represents an style metric measuring the amounts of artifacts presents in audio. Also note that SAR can be quite high even for an audio mixture that doesn't contain any artifact. Initial drops of SAR followed by subsequent improvements demonstrate that our proposed method attempts to learn to extract single-source components from the very early stages of training.

### G.4 Performance comparison on additional datasets

We have also conducted additional comparisons between the proposed method and other baselines on VGGSound (Chen et al., 2020), and AudioCaps (Kim et al., 2019) datasets. Both datasets contain a large variety of sounding source categories as well as significant environmental noise types that make the single-source separation task particularly challenging. For a fair comparison, we have reproduced all the baselines under the same setting. Moreover, we have replaced the vanilla U-Net with our improved U-Net in most baselines.

**Comparisons on VGGSound:** We have conducted performance comparison on VGGSound dataset under supervised and unsupervised with synthetic, 2- & 3-source mixtures training scenarios, as shown in Table 10. We observe unconditional methods suffer from convergence issues during unsupervised training that results in significant performance drops. Conditional methods, on the other hand, achieve considerably higher performance in general compared to their unconditional counterparts. Since the dataset contains variable length of sounding events with large amount of noise components, image-conditional methods in general achieves lower SDR compared to language-conditional methods. However, we observe our method achieves 88%, 68% of the su-

Table 10: Performance comparison on **VGGSound Dataset** under supervised and unsupervised training scenarios. Same test set of 2-Source separation is used for all cases. All methods are reproduced under the same setting. * denotes implementation with our improved U-Net model. Our proposed method largely closes the performance gap between supervised and unsupervised settings. **Bold** and blue represents the best and second best performance in each group, respectively.

| Method | Single-Source (Supervised) | Multi-Source (Unsupervised) | |
| --- | --- | --- | --- |
| | | **2-Source** | **3-Source** |
| **Unconditional** | | | |
| PIT* (Yu et al., 2017) | $2.1 \pm 0.33$ | - | - |
| MixIT (Wisdom et al., 2020) | - | $-1.7 \pm 0.44$ | $-2.9 \pm 0.51$ |
| MixPIT (Karamatlı & Kırbız, 2022) | - | $-1.4 \pm 0.51$ | $-3.1 \pm 0.39$ |
| **Image Conditional** | | | |
| CLIPSep-Img (Dong et al., 2022) | $1.3 \pm 0.34$ | $-0.5 \pm 0.27$ | $-1.2 \pm 0.35$ |
| CLIPSep-Img* (Dong et al., 2022) | $1.7 \pm 0.36$ | $0.4 \pm 0.31$ | $-0.6 \pm 0.28$ |
| SOP* (Zhao et al., 2018) | $1.6 \pm 0.23$ | $0.3 \pm 0.41$ | $-0.9 \pm 0.26$ |
| **Language Conditional** | | | |
| CLIPSep-Text (Dong et al., 2022) | $2.1 \pm 0.26$ | $0.8 \pm 0.31$ | $-0.1 \pm 0.27$ |
| CLIPSep-Text* (Dong et al., 2022) | $\mathbf{2.5} \pm 0.29$ | $1.2 \pm 0.44$ | $0.5 \pm 0.38$ |
| BertSep* | $2.0 \pm 0.27$ | $0.7 \pm 0.31$ | $0.3 \pm 0.22$ |
| CLAPSep* | $2.3 \pm 0.32$ | $1.1 \pm 0.36$ | $0.5 \pm 0.28$ |
| LASS-Net (Liu et al., 2022) | $2.2 \pm 0.31$ | $0.9 \pm 0.28$ | $0.2 \pm 0.29$ |
| Weak-Sup (Pishdadian et al., 2020) | - | $0.6 \pm 0.39$ | $-0.8 \pm 0.33$ |
| **Proposed Framework** | - | $\mathbf{2.2} \pm 0.35$ | $\mathbf{1.7} \pm 0.39$ |

Table 11: Performance comparison on **AudioCaps Dataset** representing natural multi-source mixture training. Same test set of 2-Mixture separation is used for all cases. All methods are reproduced under the same setting. * denotes implementation with our improved U-Net model. Our proposed method significantly improves the performance over the baselines. **Bold** and blue represents the best and second best performance in each group, respectively.

| Method | Test SDR |
| --- | --- |
| **Image Conditional** | |
| CLIPSep-Image (Dong et al., 2022) | $-0.7 \pm 0.47$ |
| CLIPSep-Image* (Dong et al., 2022) | $0.4 \pm 0.33$ |
| SOP* (Zhao et al., 2018) | $0.2 \pm 0.25$ |
| **Language Conditional** | |
| CLIPSep-Text (Dong et al., 2022) | $0.7 \pm 0.36$ |
| CLIPSep-Text* (Dong et al., 2022) | $1.3 \pm 0.31$ |
| BertSep* | $0.9 \pm 0.29$ |
| CLAPSep* | $1.2 \pm 0.41$ |
| LASS-Net (Liu et al., 2022) | $0.8 \pm 0.38$ |
| **Proposed Framework** | $\mathbf{2.9} \pm 0.35$ |

Table 12: SDR comparison on **3-source separation test set** form the MUSIC Dataset under supervised and unsupervised training scenarios. All methods are reproduced under the same setting. * denotes implementation with our improved U-Net model. **Bold** and blue represents the best and second best performance in each group, respectively.

| Method | Single-Source (Supervised) | Multi-Source (Unsupervised) | | |
|---|---|---|---|---|
| | | 2-Source | 3-Source | 4-Source |
| **Unconditional** | | | | |
| PIT* (Yu et al., 2017) | 2.3 ± 0.26 | - | - | - |
| MixIT (Wisdom et al., 2020) | - | -2.3 ± 0.34 | -3.1 ± 0.57 | -4.2 ± 0.35 |
| MixPIT (Karamatlı & Kırbız, 2022) | - | -1.9 ± 0.46 | -2.8 ± 0.41 | -3.9 ± 0.35 |
| **Image Conditional** | | | | |
| CLIPSep-Img (Dong et al., 2022) | 0.7 ± 0.25 | -0.8 ± 0.27 | -1.7 ± 0.35 | -2.9 ± 0.32 |
| CLIPSep-Img* (Dong et al., 2022) | 1.6 ± 0.22 | 0.1 ± 0.31 | -0.9 ± 0.28 | -1.8 ± 0.43 |
| CoSep* (Gao & Grauman, 2019) | 1.8 ± 0.28 | 0.4 ± 0.37 | -0.2 ± 0.29 | -0.7 ± 0.36 |
| SOP* (Zhao et al., 2018) | 1.3 ± 0.23 | -0.5 ± 0.41 | -1.6 ± 0.26 | -2.6 ± 0.42 |
| **Language Conditional** | | | | |
| CLIPSep-Text (Dong et al., 2022) | 1.8 ± 0.21 | -0.2 ± 0.35 | -1.1 ± 0.27 | -2.1 ± 0.45 |
| CLIPSep-Text* (Dong et al., 2022) | **2.4** ± 0.27 | 0.9 ± 0.41 | 0.3 ± 0.32 | -0.4 ± 0.41 |
| BertSep* | 1.9 ± 0.27 | 0.4 ± 0.31 | -0.2 ± 0.22 | -1.1 ± 0.27 |
| CLAPSep* | 2.2 ± 0.31 | 0.6 ± 0.36 | 0.1 ± 0.28 | -0.8 ± 0.33 |
| LASS-Net (Liu et al., 2022) | 2.1 ± 0.25 | 0.5 ± 0.26 | 0.3 ± 0.29 | -0.9 ± 0.36 |
| Weak-Sup (Pishdadian et al., 2020) | - | -1.1 ± 0.47 | -2.3 ± 0.38 | -3.2 ± 0.33 |
| **Proposed** | - | **3.5** ± 0.35 | **2.7** ± 0.42 | **2.2** ± 0.38 |

pervised method's performance on 2-source separation test set in 2-source and 3-source training scenarios, respectively. Furthermore, we achieve 1.83x, and 3.4x SDR improvements over the second best method in 2-source and 3-source training scenarios, respectively, while using the same model architecture.

**Comparisons on AudioCaps:** AudioCaps contains natural mixtures with $1 \sim 6$ single source components in each mixture which makes the separation task particularly challenging. To test on AudioCaps, we prepare a synthetic mixture-of-mixtures (MoM) test set by mixing random mixture pairs. Table 11 shows the comparison results. Due to severe convergence issues in unconditional methods, we only present comparisons for image and language conditional methods. Since the audio contains variable number of sounding sources with different durations, it becomes increasingly difficult to condition with images compared to text prompts which results in lower SDR for image conditional baselines. Nonetheless, our proposed method achieves superior performance outperforming the second highest baseline by achieving 2.3x SDR improvement under the same setting.

### G.5 COMPARISONS ON HIGHER ORDER MIXTURE TEST SETS

So far, we have shown all performance comparisons on a common test set of 2-source mixtures. Here, we present the same comparisons on more challenging test mixtures containing combinations of three single source components from the MUSIC dataset. The results are reported in Table 12. In general, we observe significant performance drops compared to 2-source test setting for all methods including ours. In particular, the mix-and-separate-based baselines suffer from $62.5\%$ SDR drop in the supervised setting in the 2-source mixture training scenario. Interestingly, our weakly supervised approach outperforms the supervised method achieving 1.5x and 1.1x higher SDR on the 3-source test set when trained on 2-source, and 3-source mixtures, respectively. This result reveals a key feature of our framework that is consistent with our observations through our other ablation studies; namely, the weak supervision proposed in our framework acts as an effective regularization mechanism, that can significantly improve the model's generalization, especially when we test it on the 3-source mixture set. In other words, the supervised method tends to overfit to the separation task on its training distribution, and that is why it experiences larger performance drop when the test distribution shifts. Whereas, in our framework, due to the inherent regularization properties of the weak supervision mechanism, the performance drop is less dramatic when the test distribution shifts.

Table 13: Ablation on the effect of CLAP-constraint in supervised single-source training. Same test set of 2-Source separation is used for all cases. All methods are reproduced under the same setting. * denotes implementation with our improved U-Net model. Bi-modal semantic CLAP constraint introduces additional regularization in supervised training, which results in notable performance improvement. **Bold** and blue represents the best and second best performance in each group, respectively.

| Method | MUSIC Dataset | | VGGSound Dataset | |
|---|---|---|---|---|
| | w/o CLAP | w/ CLAP | w/o CLAP | w/ CLAP |
| CLIPSep-Text (Dong et al., 2022) | $7.7 \pm 0.21$ | $8.2 \pm 0.32$ | $2.1 \pm 0.26$ | $2.5 \pm 0.31$ |
| CLIPSep-Text* (Dong et al., 2022) | $\textbf{8.3} \pm 0.27$ | $\textbf{8.8} \pm 0.41$ | $\textbf{2.5} \pm 0.29$ | $\textbf{2.9} \pm 0.44$ |
| BertSep* | $7.9 \pm 0.27$ | $8.3 \pm 0.35$ | $2.0 \pm 0.27$ | $2.5 \pm 0.31$ |
| CLAPSep* | $8.1 \pm 0.31$ | $8.7 \pm 0.34$ | $2.3 \pm 0.32$ | $2.8 \pm 0.31$ |

## G.6 EFFECT OF BI-MODAL CLAP CONSTRAINT ON SUPERVISED TRAINING

Apart from using bi-modal CLAP constraint as weak-supervision for multi-source (unsupervised) training, we study its impact on single source (supervised) training on the baseline methods. In particular, we add the bi-modal semantic CLAP-constraint in the form $\mathcal{L}_{CNT}$ loss to the mix-and-separate $\mathcal{L}_{URL}$ loss, while training using supervised single-source samples from the MUSIC and VGGSound datasets. The results are reported in Table 13. As the results show, there is a consistent SDR improvement across the board when we incorporate the CLAP constraint in supervised learning, even though, intuitively speaking, the weak supervision obtained from $\mathcal{L}_{CNT}$ should be impertinent in the presence of the strong supervision signal coming from the supervised loss. We hypothesize that the integration of CLAP constraint here introduces additional regularization to supervised training by transferring the knowledge obtained through CLAP's large-scale pre-training to the problem of audio source separation. This result further shows that our proposed framework not only boost the separation quality in unsupervised and semi-supervised training scenarios, it can also help the supervised training itself by introducing extra cross-domain regularization.

## G.7 ADDITIONAL COMPARISONS FOR SEMI-SUPERVISED TRAINING

Table 14 depicts additional performance comparisons between supervised training on single source sounds, unsupervised training on multi-source mixture sounds, and proposed semi-supervised training on both single-source and multi-source mixture sounds. We split the MUSIC training dataset with different ratios for single-source and multi-source training as mentioned before. Multi-source mixtures are composed of two single-source components here. In general, semi-supervised training significantly outperforms both supervised and unsupervised training. With the increase in single source data portion, we note the performance improves in general. Similarly, unsupervised performance on multi-source mixtures also depend on available training data. Also note that the unsupervised performance with different splits of training data largely closes the performance gap in comparison with single-source supervised training, which is consistent with our prior observations. More notably, however, by combining both single-source and multi-source training mixtures in the proposed semi-supervised learning framework, we achieve considerable performance improvement compared to $100\%$ supervised performance reaching $9.5$ SDR, which is $28\%$ higher than the $100\%$ scenario for the supervised baseline. This result, again, suggests the regularization effects of the proposed framework which can significantly reduce the reliance on single-source data and supervised training for conditional sound separation.

## G.8 EFFECTS OF PROMPT TUNING FOR THE CLAP MODEL

We primarily use the bi-modal CLAP model to generate weak supervision for single-source separation from the corresponding language entity. Since the CLAP model is trained on large corpora of audio-language pairs, it can effectively generate weak supervision signals for a target dataset based on hand-crafted language prompts. We have studied the performance impacts of the CLAP model customization on a target dataset by tuning language prompts with few-shot single-source reference audio samples. The results are given in Table 15. For this experiment, first we separate

Table 14: Performance comparisons between supervised, unsupervised, and semi-supervised settings using both single source and multi source (2-source) training data from the MUSIC dataset. Test SDR on 2-source mixtures is reported. $x\%$ of training data is used for single-source supervised training, while $(1 - x)\%$ of data is used for multi-source weakly-supervised training with synthetic mixtures. CLAPSep is used for supervised training, while our proposed weakly supervised training is applied for the multi-source data. Lastly, the semi-supervised training is applied on the combination of single source and multi-source data. Semi-supervised learning consistently achieves better performance in all data splits. Training on single-source, multi-source, and joint single- and multi-source data are referred as "Supervised", "Unsupervised", and "Semi-supervised" method, respectively. **Bold** and blue represents the best and second best performance in each group, respectively.

| Single Source Split | Two Source Split | SDR Performance | | |
|---|---|---|---|---|
| | | Supervised | Unsupervised | Semi-Supervised |
| - | 100% | - | **7.9** | - |
| 5% | 95% | 2.6 | 7.6 | 8.8 |
| 10% | 90% | 3.9 | 7.4 | 8.9 |
| 25% | 75% | 5.3 | 7.1 | 9.2 |
| 50% | 50% | 6.6 | 6.2 | 9.4 |
| 75% | 25% | 7.4 | 4.9 | **9.5** |
| 100% | - | **8.1** | - | - |

Table 15: The effects of prompt tuning for our proposed framework. We use the 2-source training mixtures from MUSIC dataset. Test SDR on 2-source mixtures is reported. We initially separate several full length single source audio samples from each category for training learnable prompts. **Bold** and blue represents the best and second best performance in each group, respectively.

(a) Ablation study on learnable prompt length with number of training audio samples per category. Each full-length audio represents a single source audio. We extract several overlapping frames from audio samples to train the learnable text prompts.

| Prompt Length | #Audios/Category | Test SDR |
|---|---|---|
| | 1 | 8.1 |
| 8 | 2 | 8.4 |
| | 5 | 8.6 |
| | 1 | 8.2 |
| 16 | 2 | 8.5 |
| | 5 | 8.7 |
| | 1 | 8.0 |
| 32 | 2 | 8.6 |
| | 5 | **8.8** |

(b) Comparison between the learnable and the template-based prompts.

| Prompt Type | Test SDR |
|---|---|
| Template-based | 7.9 |
| Learnable | 8.8 |
| OCT (Tzinis et al., 2023) | 8.7 |
| OCT++ (Tzinis et al., 2023) | **9.0** |

Table 16: Subjective evaluation performance analysis. * denotes implementation with our improved U-Net model. **Bold** and blue represent best and second best performance, respectively.

| Training Data | Method | Correct (%) ↑ | Wrong (%) ↓ | Both (%) ↓ | None (%) ↓ |
|---|---|---|---|---|---|
| Supervised | CLIPSep-Text* (Dong et al., 2022) | 71.1 | 0.9 | 26.9 | 1.1 |
| Unsupervised | CLIPSep-Text* (Dong et al., 2022) | 30.4 | 20.5 | 40.6 | 8.5 |
| | Proposed Framework | 68.9 | 1.5 | 27.4 | 2.2 |
| Semi-supervised | Proposed Framework | **82.6** | 0.4 | 16.2 | 0.8 |

few samples of full-length single-source audio samples for each category to incorporate learnable language prompts instead of the template-based ones. We then randomly sample single-source audio segments from a hold-out dataset to train learnable language prompts. By learning such prompts, we can customize the CLAP model for our target dataset to generate more informative supervision signal. In Table 15a, we report the effects of the prompt lengths as well as the number of full-length audio samples per category on the 2-Source test set using the proposed weakly supervised training on 2-source mixtures. By using 5-shot single-source audio samples per category and the prompt length of 32, we achieve around 12% SDR improvement compared to the template-based prompts (Table 15b).

Apart from the text-based prompt tuning using CLAP model, our proposed framework can also integrate heterogeneous prompting with other cues of the target source. Following Tzinis et al. (2023), we experiment with heterogeneous training conditions, such as text description, signal energy, and harmonicity of the target sound for source separation. We use the hold-out single source samples (5/Category) for each category to estimate the additional cues for prompting target sounds in mixtures. The baseline OCT method performs on-par with our learnable text prompting technique (8.7 vs. 8.8). We note that OCT with the embedding refinement approach (OCT++) achieves the best performance of 9.0 SDR. Hence, our proposed framework can effectively integrate advanced prompting techniques to separate the target sounds from the mixture.

## H    SUBJECTIVE EVALUATION

We conduct a subjective human evaluation to compare different models' performances based on human perception. Following prior work (Zhao et al., 2018; Dong et al., 2022), we have randomly sampled separated sounds from 2-source mixtures and presented them to the evaluators, who are then asked "Which sound do you hear? 1. A, 2.B, 3. Both, or 4. None of them". Here, A and B are replaced by the single-source sounding entities present in the input mixture, *e.g.* A. cat meowing, B. pigeon, dove cooing. In Table 16, we present the percentages of predicted samples that are correctly identified by the evaluator as the source class (Correct), which are incorrectly perceived by the evaluator (Wrong), which contains audible sounds of both sources (Both), and which doesn't contain any of the target sounds (None). We use the same 30 sample predictions on 2−source mixture test sets for comparing models trained with supervised single-source, unsupervised multi-source, and semi-supervised single with multi-source data. 20 human evaluators have participated in this evaluation. We use the CLIPSep (Dong et al., 2022) method as the competitive baseline of the *mix-and-separate* framework with the text prompts.

As the results show, our proposed framework improves over the CLIPSep baseline's correct percentage statistics of 30.4% in the unsupervised setting by more than twice, reaching 68.9% and almost closing the gap with the performance of CLIPSep under the supervised regime (*i.e.* 71.1%). Furthermore, our framework's performance under the semi-supervised training setup goes even beyond that of the supervised setting by significantly reducing the number of under-separated instances from 26.9% to 16.2%, leading to 11.5% increase in correct percentage statistics to the total of 82.6%. This result shows the efficacy of our weakly-supervised training strategy under both unsupervised and semi-supervised training regimes. But more importantly, these results are consistent with our quantitative evaluation results, which further corroborate our conclusions.

# I   QUALITATIVE COMPARISONS

In this appendix, we present qualitative comparisons between the proposed method and the mix-and-separate approach represented by the CLIPSep Dong et al. (2022) framework under the unsupervised training scenario. The MUSIC dataset is used for this analysis. The models are trained and tested using 2-source mixtures. The results are given in Figure 8 - Figure 11. Due to the lack of single-source supervision in the mix-and-separate approach, most of its predictions exhibit significant spectral leakage, and large cross-interference. In contrast, our proposed method significantly reduces the spectral loss and cross interference. Also, separation under challenging cases of spectral overlap produces reasonable performance. These examples demonstrate the effectiveness of the proposed weakly supervised training method in disentangling single-source audio components form the input mixture.

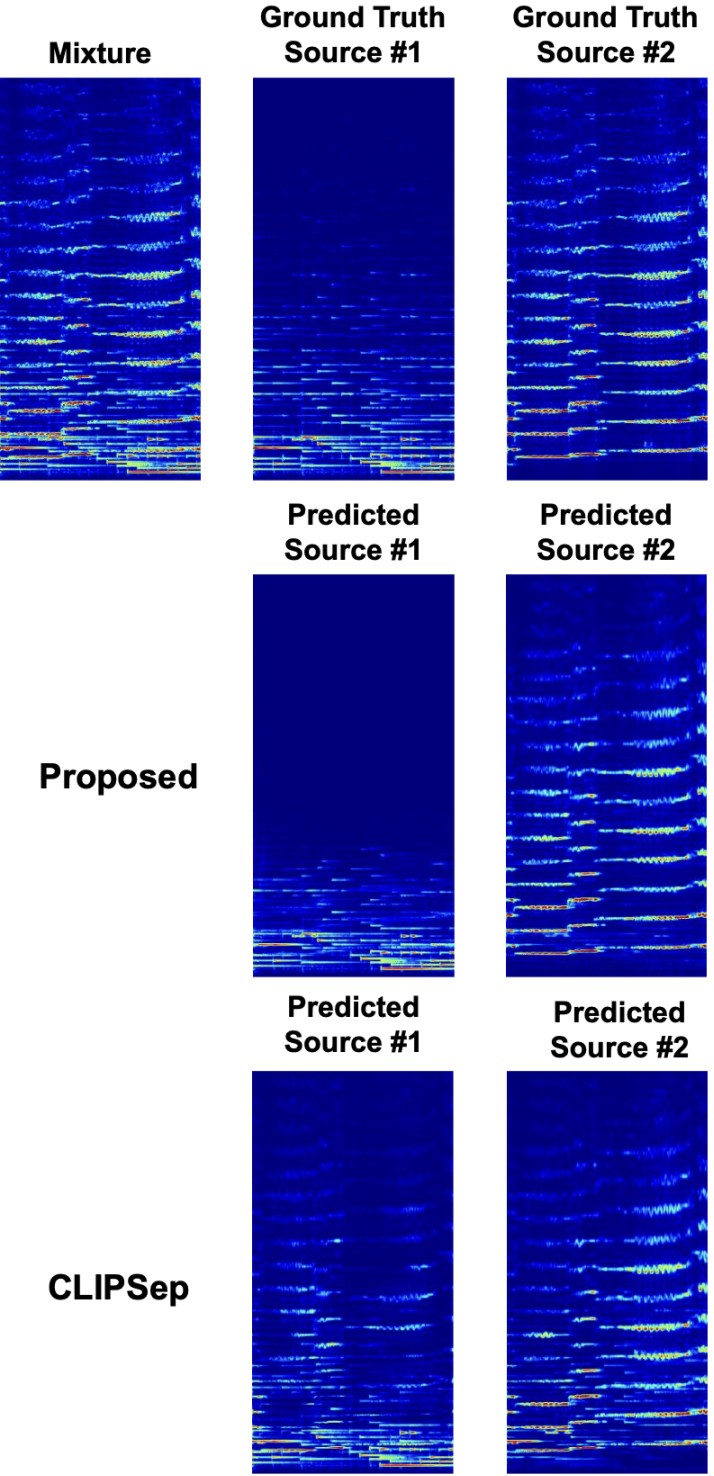

Figure 8: Qualitative comparisons between the proposed method and the mix-and-separate approach (CLIPSep (Dong et al., 2022)): The input mixture contains *piano* (source 1) and *violin*(source 2) sounds. For the lack of single source supervision in CLIPSep, large cross-interference is visible in its prediction for the *piano* source . In contrast, our method significantly reduces cross-interference.

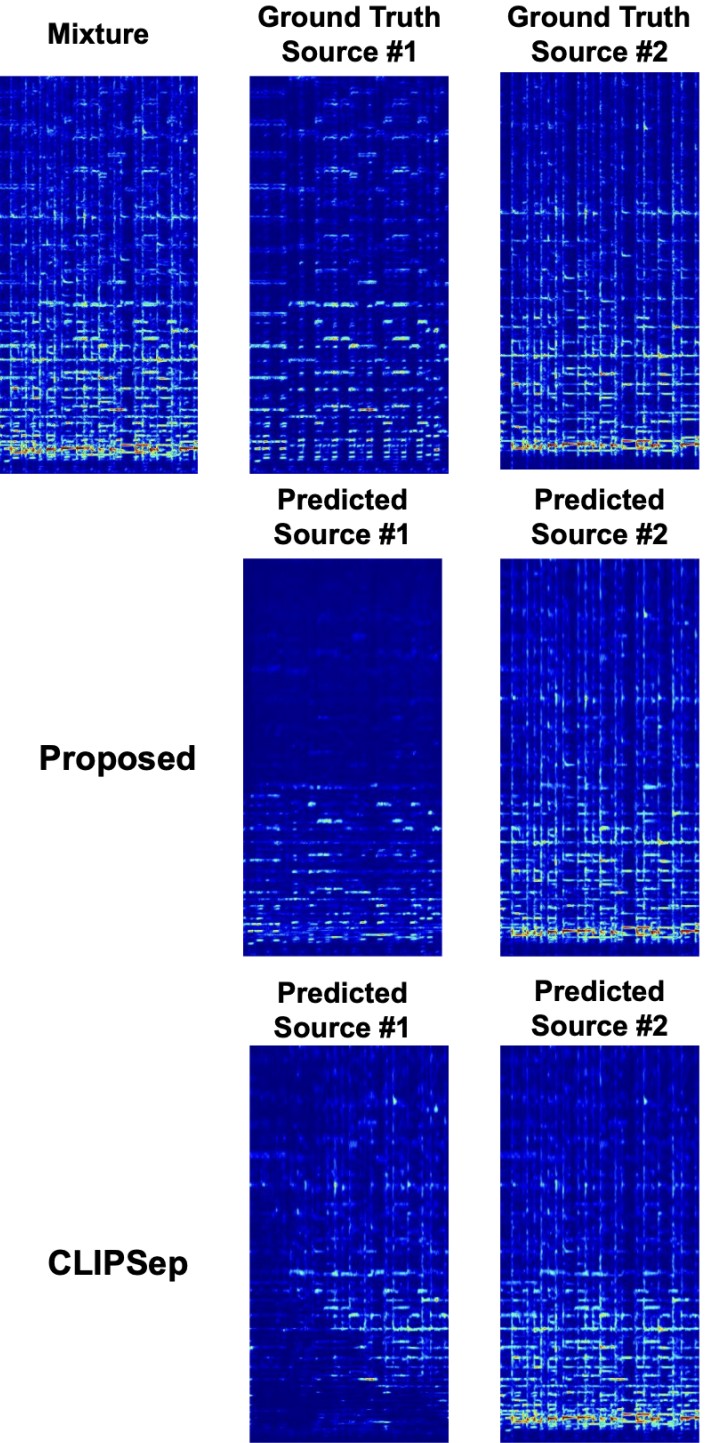

Figure 9: Qualitative comparisons between the proposed method and the mix-and-separate approach (CLIPSep (Dong et al., 2022)): The input mixture contains *accordion* (source 1) and *ukulele*(source 2) sounds. For *accordion* sound separation, CLIPSep exhibits significant spectral loss, while for *ukulele* sound separation, it shows cross-interference. However, our method largely reduces both the spectral loss for *accordion* sound segmentation and the cross-interference for *ukulele* sound segmentation.

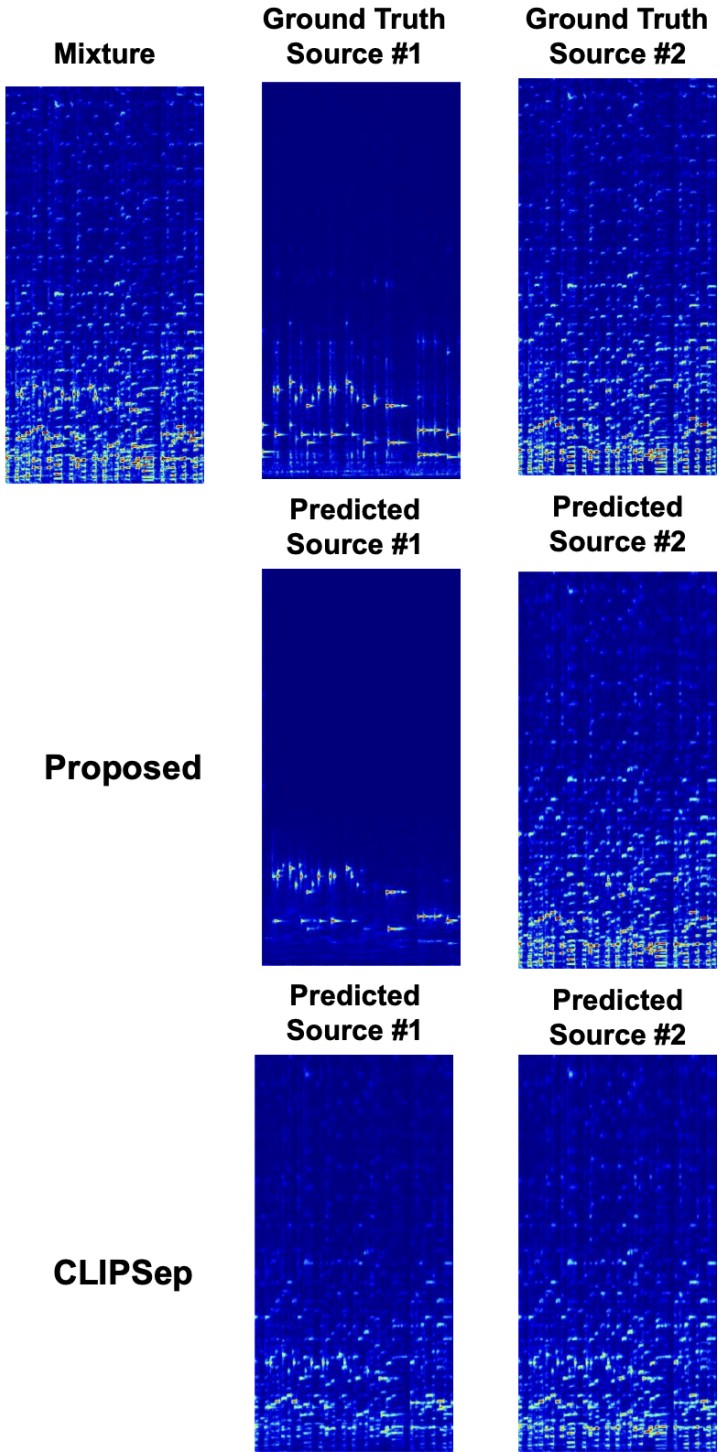

Figure 10: Qualitative comparisons between the proposed method and the mix-and-separate approach (CLIPSep (Dong et al., 2022)): The input mixture contains *xylophone* (source 1) and *accordion*(source 2) sounds. CLIPSep can hardly differentiate between these two sources due to a large spectral overlap. Our method, however, reasonably separates the two sounds despite showing some spectral loss for the *accordion* prediction.

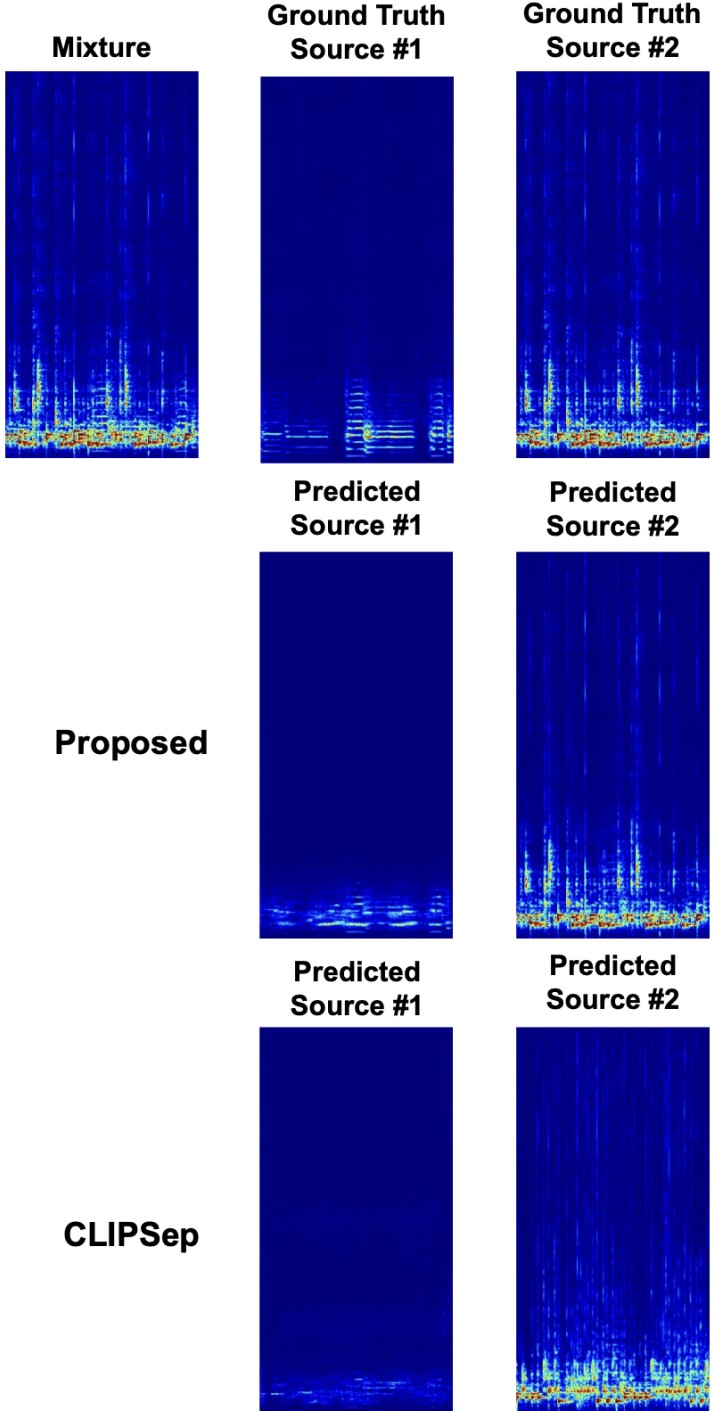

Figure 11: Qualitative comparisons between the proposed method and the mix-and-separate approach (CLIPSep (Dong et al., 2022)): The input mixture contains *tuba* (source 1) and *congas*(source 2) sounds. CLIPSep cannot properly identify the *tuba* sound due to the lack of single source training. Whereas, our method achieves considerable results in separating the *tuba* sound. nevertheless, some spectral loss can be observed for the *tuba* sound separation, which is reasonable considering the significant spectral overlaps of the two sounds.

