# OpenReview forum: "Weakly-supervised Audio Separation via Bi-modal Semantic Similarity"
_ICLR.cc/2024/Conference — ICLR 2024 poster_

### Official Review · Reviewer_fPmw · 2023-10-31

**Soundness:** 3 good
**Presentation:** 3 good
**Contribution:** 3 good
**Rating:** 8
**Confidence:** 3

**Summary:**

The paper presents a framework designed to utilize text-audio data pairs, utilizing the text-audio encoding network, CLAP, for source separation. There are two primary training objectives:

1. Unsupervised Training: Two mixed audio recordings are combined, and the separation network aims to reconstruct each original mixture.
Weakly-supervised
2. Weakly-supervised Audio-language Training: Here again, two mixed audios are combined. The goal is to separate each individual source present in both mixtures. This is achieved with the bi-modal contrastive loss and a reconstruction loss. The bi-modal contrastive loss is employed to enhance the similarity between the predicted single source and its corresponding language prompt. The reconstruction loss, mirroring the unsupervised training, aims to reconstruct the original individual mixture prior to combining.
The paper provides comprehensive experiments, demonstrating that the proposed method surpasses baseline methods in performance.

**Strengths:**

1. The paper introduces a unique framework that integrates unsupervised, weakly supervised, and semi-supervised training for source separation, conditional on language prompts. The experimental results validate the efficacy of the proposed method.
2. The paper conducts detailed experiments, contrasting the proposed approach with various baselines. Ablation tests further underscore the superior performance of the proposed method compared to existing techniques.

**Weaknesses:**

Although it might not be a weakness, it is a bit questionable how the model performance will improve in large-dataset settings.

**Questions:**

Does the proposed method necessitate knowledge of all components present in the mixture? Furthermore, if the data contains background noises not mentioned in the text prompt, could this impact the model's performance? In a weakly-supervised setting, it's assumed that the individual sources contribute to the mix. For instance, if the prompt mentions people talking and birds chirping, but the recording was made in a city environment with background motor sounds, how would the model handle such discrepancies?

---

> ### Author Response · Authors · 2023-11-16
> **Our response**
>
> We appreciate the reviewer’s positive review of our work and the interesting questions raised. In the following, we will address these questions.
>
> > 1) "Although it might not be a weakness, it is a bit questionable how the model performance will improve in large-dataset settings."
>
> **Our response:** in our experiments, the VGGSound and AudioCaps datasets contain nearly 200K and 50K samples, respectively. In addition to extensive experimentation on each dataset (Table 1, 8, 10), we present large-scale semi-supervised training across datasets in Table 2. Therefore, in that sense, we can confidently claim our proposed framework yields significant improvement under decent size data regimes. However, if the reviewer’s question is to whether the gain that we obtain through our proposed weakly-supervised framework starts diminishing as the capacity of the model and the training data size becomes very large, that is an interesting research question. Intuitively speaking, that should happen when the generalization error starts converging toward its minimal value (i.e. the inherent noise/uncertainty level for the problem) as the model capacity and training data size become larger and larger. Answering this question experimentally, however, is beyond the scope of this work. Nonetheless, we believe this should not be considered as a weakness of our work since this issue can be equally raised against any contribution in the field aiming at reducing the generalization error of a Machine Learning model. And while the diminishing return phenomenon might indeed occur in massive-scale training regimes, we believe it is still valuable to significantly improve the model generalization in medium-scale training settings, which are, by the way, more accessible to the majority of researchers and practitioners in the field.
>
> > 2) “Does the proposed method necessitate knowledge of all components present in the mixture? Furthermore, if the data contains background noises not mentioned in the text prompt, could this impact the model's performance?”
>
> **Our response:** having the knowledge of all components in the mixtures would be ideal; however, realistically speaking, that’s not always possible; in such scenarios, the absent sources from the textual captions effectively act as “noise” during training. This type of noise particularly misleads the Consistency Reconstruction Loss (CRL) component which aims at reconstructing the original mixture from the predicted single-source components. But the good news is that depending on the noise level of our training dataset we can contain its negative impact by adjusting the relative weight of the CRL component. In fact, we have conducted ablation studies to find the optimal weight ratios for different loss components using heldout data from each dataset. The results are added in Table 7 and Appendix G.2 in the revised version. Interestingly, we observe that the optimal relative weight of CRL drops more than 50% from MUSIC dataset to VGGSound dataset, and 50% from VGGSound dataset to AudioCaps dataset. This trend happens to be the same order as the noise-level order for these datasets, which suggests that using this procedure, we can adapt the training dynamics of our framework based on the noise-level of the training dataset to minimize the negative effects of noise in training data.
>
> > 3) “… if the prompt mentions people talking and birds chirping, but the recording was made in a city environment with background motor sounds, how would the model handle such discrepancies?”
>
> **Our response:** Such discrepancies are partly handled by the CNT loss component which incorporates the frozen CLAP model pretrained on the large-scale text-audio pairs. The implicit assumption here is that the existence of extraneous and noise components in the model’s prediction would negatively impact the CLAP similarity score, which is reasonable due to large-scale pre-training of CLAP.  As mentioned earlier, the other way we address this issue is by adjusting the relative weight of the CRL loss component based on the dataset noise level by conducting a hyper-parameter search procedure before training (see Appendix G.2). By doing so, we effectively minimize the negative impacts of the extraneous and noise components in the training data.
>
> For further demonstration, we invite the reviewer to check out the new qualitative demo website uploaded on the anonymous Google Drive link below, containing samples with background noise and/or extraneous source components and their separation results using our model and the CLIPSep baseline.
>
> Qualitative Demo: https://drive.google.com/file/d/1tR7o8cXoHqS9Ns8riHB_TD6Hfenoz0zH/view

---

> > ### Comment · Reviewer_fPmw · 2023-11-22
> > **Thanks for the reply**
> >
> > Thank you for the reply. Thanks for the additional experiments and results which, to some extent, answers my questions.

---

> > > ### Author Response · Authors · 2023-11-22
> > > **Appreciating Your Acknowledgment**
> > >
> > > We appreciate your recognition of our clarifications and the promptness of your response. Your thorough review and invaluable feedback throughout this process are deeply valued.

---

### Official Review · Reviewer_SLiU · 2023-11-01

**Soundness:** 2 fair
**Presentation:** 3 good
**Contribution:** 3 good
**Rating:** 6
**Confidence:** 5

**Summary:**

This paper introduces a novel conditional source separation model designed to improve separation performance in weakly-labeled data. The approach leverages CLAP embeddings to enforce semantic consistency in the separation results, even when individual source tracks are not explicitly provided but merged within the data.

The proposed model comprises three key components:

1. Mix-Separation Paradigm: This component creates separation datasets from weakly-labeled noisy data, where more than one source is present in a single audio clip.

2. CLAP Embeddings: The model employs CLAP, a joint embedding of text and audio modalities, to establish a constraint that ensures the separation results align with the same semantic characteristics, primarily focusing on timbre. This constraint is enforced through the CLIPLoss between the separation audio embeddings and the associated single-source text embeddings (e.g., 'This is the sound of [class]').

3. Training: The entire separation model is trained by integrating the above two components with corresponding loss functions.

The experimental results demonstrate that the incorporation of CLAP embeddings significantly enhances separation performance by making the model sensitive to mixed and overlapping events in the data, ultimately leading to superior separation outcomes.

**Strengths:**

The paper presents a compelling and well-articulated examination of a critical limitation within the existing zero-shot/few-shot conditional source separation pipeline: the dataset. It addresses this issue within the context of cross-modal learning (CLAP).

The paper's technical novelty is commendable. It ingeniously employs CLAP to establish a method for validating the consistency of separation outputs, integrating loss objectives that effectively converge both intermediate semantic consistency and separation content accuracy. Furthermore, it extends this innovative approach into a semi-supervised learning framework, broadening the potential applications of the data.

The paper demonstrates a robust and comprehensive experimental performance evaluation. It goes beyond merely listing state-of-the-art results by including self-implemented ablations, which, while not previously explored, offer valuable points of comparison. The extensive analysis of existing work enriches the paper's conclusion and strengthens its contribution to the field

**Weaknesses:**

I assigned a 'weak reject' score to this paper due to what I consider to be a critical issue in the design of the separation pipeline and an empirical problem in the experimental results.

Firstly, the paper introduces the utilization of CLAP embeddings as a constraint to enforce semantic similarity between the separation output and the source constraint. While this is a technically novel approach, it raises two important concerns:

(1) The use of CLAP, or similar contrastive learning models, to enforce semantic similarity is not as solid as it appears. This is because CLAP is also trained with the same noisy, weakly-labeled data. When the paper uses prompts like '[this is the sound of flute],' these prompts have been trained on audio clips that contain more than just flute sounds, potentially including piano and guitar sounds. Therefore, the data used in the current setting for training this proposed separation model, is exactly the data that CLAP was trained on (and even more). This raises questions about whether the constraint effectively operates as intended, especially in cases where the separation sample contains multiple sources, one of which is the target source. To ascertain the true effectiveness of CLAP as an exclusive text embedding, further experiments are necessary.

(2) The semantic alignment is a fancy phrase but in this audio separation case, primarily restricts the timbre of the audio rather than aligning the semantic content. While a high CLAP score indicates similarity in timbre, it doesn't guarantee similarity in content. Semantic alignment comes more in the case when the text itself contains different arrangement and logics. However, here the model only uses one or two single prompt such as [this is the mixture/sound of [class]]. A worth-noting question to ask is: why not use a timbre classier to verify it? You can send the separation result to an audio classification model to force the output to be in one timbre. Or a more direct question is: did you try this as an ablation study to show that using CLAP to align the semantic similarity is better than training a timbre classifier to enhance the separation results? I think this ablation study is a good point to show the technical solid of this paper. Such an ablation study would strengthen the paper's technical foundation without compromising its flexibility. Because for every data trained in this pipeline, you should at least know the source type to make the prompt text. A timbre classifier seems can be trained in a fixed (but large) candidacy slot.

From another perspective, the paper presents a substantial number of experiments to demonstrate the efficacy of the separation results. The appendix is comprehensive and aids in clarifying various aspects. However, the experimental results on the MUSIC dataset, achieving an SDR performance of 7.9 and 9.9 under the semi-supervised learning paradigm, do not significantly outperform the results in the recent LASSNet paper [1] (9.75 dB and 10.45 dB). While I acknowledge that both papers were produced during the same time period, this raises concerns about the necessity of CLAP as an embedding for enforcing semantic similarity. It prompts a reconsideration of whether CLAP could be superseded by a simpler text- or audio-query model with CLAP or CLIP embeddings, but a different separation design that doesn't require semantic similarity.

Besides, any listenable demo for this paper? It would be beneficial to provide a demo page to assess the performance of the separation and we might see more interesting findings, such as how CLAP is trying to enhance the separation result.

In light of these concerns, I eagerly anticipate the authors' response to address these questions. Until it can be convincingly demonstrated that CLAP is a crucial element in enhancing performance or that it offers greater functionality than a single classifier constraint, I believe this paper may fall slightly below the acceptance threshold for ICLR.


[1] Separate Anything You Describe, Arxiv.

**Questions:**

Two questions are proposed in the weaknesses section, regarding the question of CLAP vs. timbre classifier, and the latest result of LASSNet either using CLAP or CLIP as the query embedding.

---

> ### Author Response · Authors · 2023-11-16
> **Our response**
>
> We appreciate the reviewer’s diligent review and helpful suggestions. We have followed the reviewer’s suggestions to address their concerns, as we will explain below:
>
> > 1) "The use of CLAP [...] is not as solid as it appears. This is because CLAP is also trained with the same noisy, weakly-labeled data. [...] This raises questions about whether the constraint effectively operates as intended,.."
>
> **Our response:** we completely agree with the reviewer that the CLAP model does not provide sufficient supervision for the audio separation task on its own, not just because it’s trained on noisy data but also because the bi-modal semantic alignment itself is a different task than audio separation. This is exactly why we use the CLAP loss in combination with the mix-and-separate and consistency losses (see Table 6). Nevertheless, since CLAP has been trained on large-scale language and audio pairs, it provides an informative regularization signal for the separation task while at the same time enabling the introduction of single-source prompts via weak supervision. By incorporating it within our framework, we effectively transfer that knowledge to the audio source separation domain. To further show the importance of the CLAP loss, we have conducted a new set of ablation studies where we add the CLAP loss to the purely supervised learning; the results are added under the new appendix section G.6 and Table 11 in the revised version. As these results show, the addition of CLAP loss significantly improves the supervised learning performance across all cases.  In other words, the CLAP loss provides valuable regularization, even in the case of purely supervised learning where we have full access to single-source samples during training.
>
> > 2) "... why not use a timbre classier to verify it? [...]. Or a more direct question is: did you try this as an ablation study to show that using CLAP to align the semantic similarity is better than training a timbre classifier ..."
>
> **Our response:** we have repeated our experiments with our proposed CLAP-based loss replaced by a timbre classifier, where the classifier shared the exact same architecture as that of the CLAP audio encoder, but the contrastive loss is replaced by the cross-entropy loss. We trained this loss using the same training data for the original separation problem under two scenarios: (I) concurrently with the separation model, and (II) pretrained beforehand. See the added results in Table 1 and the discussion in Section 4.1 of the revised version for more details. In short, we have observed that the concurrent version performs worse than some of the baselines, and while the pretrained version does better than the baselines, its performance is still significantly lower than our proposed CLAP loss, not to mention its restricted applicability due to the fixed number of classes. We hypothesize the superiority of the CLAP-based supervision comes from the large-scale pretraining of CLAP which enables us to transfer that knowledge to source separation. In other words, in the limit for large-scale training data and gigantic number of classes, the classification approach should perform as well as the CLAP-based loss, but at that point, we might as well use CLAP.
>
> > 3) "... However, the experimental results on the MUSIC dataset, achieving an SDR performance of 7.9 and 9.9 under the semi-supervised learning paradigm, do not significantly outperform the results in the recent LASSNet paper [1] ...."
>
> **Our response:** please note that the cited work and ours report two different metrics (SI-SDR and SDRi vs. SDR) which are not directly comparable. But even if we both used the same metrics, these two results would not be still comparable. Note that, here our primary contribution is a novel weak-supervision training strategy to reduce the gap between the training and test data distributions in the LASS problem. Our experiments are also mainly designed to evaluate this hypothesis while controlling for other factors. This is why unlike many other existing works (including AudioSep [1]), we've insisted on re-implementing the competitor methods using the same architecture and training data as much as possible. With that in mind, an AudioSep model trained using the same datasets and architecture backbone as in our experiments boils down to the “Supervised” row in our Table 2. This is because an AudioSep model is trained using the vanilla supervised strategy, for which we have shown the addition of our weak supervision through the CLAP loss does indeed improve the SDR significantly from 8.1 to 9.5. Of course if, like AudioSep, we had also trained our model on a much bigger multitude of datasets, the SDR could have been even higher, but that was not our primary objective in this work.
>
> > 4) "any listenable demo for this paper? ..."
>
> **Our response:** we have prepared a new qualitative demo website uploaded here: https://drive.google.com/file/d/1tR7o8cXoHqS9Ns8riHB_TD6Hfenoz0zH/view

---

> > ### Author Response · Authors · 2023-11-22
> > **Your feedback is appreciated**
> >
> > Dear Reviewer SLiU,
> >
> > Since the the rebuttal period ends today, we'd like to know if you have any questions regarding our responses and the new additions to the submission so we can try to address it before the window closes. Therefore, we appreciate your feedback.
> >
> > Sincerely,
> > The authors

---

> > ### Comment · Reviewer_SLiU · 2023-11-23
> > **Response to the Authors' Rebuttal**
> >
> > I really appreciate the authors' insightful and throughout response, which addresses me a lot of questions.
> >
> > After carefully reading the reply, I would like to raise my score as 6.
> >
> > This paper meets my expectation to the acceptance of ICLR.

---

> > > ### Author Response · Authors · 2023-11-23
> > > **Gratitude for Your Acknowledgment**
> > >
> > > We appreciate your recognition of our clarifications and your openness to updating the score. We greatly value your thorough review and valuable feedback throughout this process.

---

### Official Review · Reviewer_ScdR · 2023-11-01

**Soundness:** 2 fair
**Presentation:** 3 good
**Contribution:** 2 fair
**Rating:** 6
**Confidence:** 4

**Summary:**

In this paper, a novel weakly supervised learning framework is introduced for language-conditioned audio separation, particularly when single-source audio samples are absent during training. Leveraging the cross-modal similarity between audio and language using the pretrained CLAP model, this approach generates weak supervision signals for single-source prompts during training, effectively bridging the gap between training and test data distributions. The authors perform experiments to showcase the superiority of this framework over other baseline models and they show that they can bridge the gap between the performance of the unsupervised and the supervised methods.

**Strengths:**

- The paper is generally speaking well written with a lot of information needed to be reproduced by the researchers in the field of sound separation.
- The authors have conducted several experiments to show that their algorithm performs better than some other existing text-based separation algorithms.
- It is very important that the proposed method yields a significant performance improvement over the unsupervised baseline of MixIT.
- I also like the novel extension of MixIT with conditioning.

**Weaknesses:**

1. The authors should make clear the distinction of when the proposed method is trained using only weak supervision and when it is semi-supervised trained. For instance, in Table 1, I think the proposed framework row refers to the semi-supervised version of the method, thus the authors should rename the column to ‘Fully supervised’ from ‘Supervised’. Maybe a better idea is to specify the data used to train ALL the parts of each model and have two big columns ‘Mixture training data’ and ‘Single source data’ which will make it much more prevalent of what is which.

2. Building upon my previous argument, I think that when one is using these large pre-trained networks on single-source data like CLAP, the underlying method becomes supervised in a sense, or to put it more specifically supervised with unpaired data. The authors should clearly explain these differences throughout the manuscript.

3. I think the authors should include stronger text-based sound separation baselines like the model and ideally the training method that uses heterogeneous conditions to train the separation model in [A] which has already shown to outperform LASS-Net (Liu et. al 2022) which is almost always the best performing baseline in this paper.

I would be more than happy to increase my score if all the above weaknesses are addressed by the authors.

[A] Tzinis, E., Wichern, G., Smaragdis, P. and Le Roux, J., 2023, June. Optimal Condition Training for Target Source Separation. In ICASSP 2023-2023 IEEE International Conference on Acoustics, Speech and Signal Processing (ICASSP) (pp. 1-5). IEEE.

**Questions:**

Would it be possible to add some human evaluations for the best baseline and the proposed method?

---

> ### Author Response · Authors · 2023-11-16
> **Our response**
>
> We appreciate the reviewer’s meticulous feedback and suggestions. We have done our best to meet the reviewer’s suggestions. Below, we discuss these efforts point by point and address the concerns raised by the reviewer.
>
> > 1) "The authors should make clear the distinction of when the proposed method is trained using only weak supervision and when it is semi-supervised trained..."
>
> **Our response:** we apologize for the lack of clarity. All the results in Table 1 including our proposed framework row are actually obtained under *unsupervised* (multi-source) training regime. The supervised (single-source) column in Table 1 is merely provided as an upper-bound for the competitor models’ performances. On the other hand, the results for semi-supervised (i.e., mix of single-source and multi-source training samples) training scenario are provided in Table 2. We appreciate the reviewer’s useful feedback about the terminology clarification; following that, we have added a paragraph in Section 4 in the revised version, describing our terminology as the reviewer suggested. We have also clarified the terminology through our tables and captions.
>
> > 2) "... I think that when one is using these large pre-trained networks on single-source data like CLAP, the underlying method becomes supervised in a sense, or to put it more specifically supervised with unpaired data..."
>
> **Our response:** we note that CLAP is NOT pretrained on single source samples for the most part to begin with. The training data for CLAP consists of large corpora of *uncurated* audio-text pairs, many of which contain more than one sounding source mixed with various background noise components while accompanied by arbitrary text captions. Therefore, based on our definition of supervised, CLAP is not a supervised model either. The only advantage of CLAP over our framework is that it has been trained on massive-scale training data, which has enabled it to discern language semantics despite the noise. Even after such massive pre-training, CLAP by itself alone is not sufficient for audio source separation and needs our other loss components (see Appendix G.2 and Table 6). Only after combining it with unsupervised reconstruction and consistency losses, we can introduce the benefit of weak supervision on the top of the classical mix-and-separate approach, but that weak supervision itself comes from large-scale, self-supervised pretraining rather than supervised learning. In that sense, our proposed framework cannot be considered a supervised technique.
>
> > 3) "I think the authors should include stronger text-based sound separation baselines like the model and ideally the training method that uses heterogeneous conditions to train the separation model in [A] ..."
>
> **Our response:** we would like to emphasize that our primary contribution in this work is a novel weakly-supervised training strategy which is orthogonal to other aspects one may employ to boost the performance of source separation models including using heterogeneous conditions. To further show this, we have originally conducted experiments using learnable text conditions (as opposed to template-based conditions) to show that the SDR for our framework can be further improved by employing learnable prompts (see Appendix G.8). Nevertheless, per the reviewer’s question, we also conducted new ablation experiments using the OCT and OCT++ methodologies proposed in Tzinis et al. 2023. The results and further discussions are added to  Appendix G.8 in the revised version. In short, we observe both OCT and OCT++ improve our framework’s performance compared to using template-based prompts, and while OCT gives us an improvement on par with the learnable prompt approach, OCT++ goes even further. But then again, these additions are completely orthogonal to what we have primarily proposed and evaluated in this work, and they can improve both the baselines and our proposed framework.
>
> > 4) "Would it be possible to add some human evaluations for the best baseline and the proposed method?"
>
> **Our response:** we have performed a subjective evaluation involving 20 human evaluators comparing our framework with CLIPSep-Text (Dong et al. 2022) which was the second best method overall under both unsupervised and semi-supervised training regimes. We have followed the methodology presented in Zhao et al., 2018 and Dong et al., 2022 for subjective evaluation. The results are added under a new appendix (Appendix H) in the revised version. As these results show, our framework significantly outperforms the baseline under both unsupervised and semi-supervised training setups according to the human evaluation. More importantly, these results are consistent with our quantitative evaluation results, which further corroborate our conclusions.
>
> Finally, for further demonstration, we invite the reviewer to check out the new qualitative demo website uploaded here: https://drive.google.com/file/d/1tR7o8cXoHqS9Ns8riHB_TD6Hfenoz0zH/view

---

> > ### Comment · Reviewer_ScdR · 2023-11-20
> > **Response 1**
> >
> > Thank you for addressing most of my comments. I will increase my score accordingly.

---

> > > ### Author Response · Authors · 2023-11-21
> > > **Thank you for your acknowledgement**
> > >
> > > Thank you for acknowledging our clarifications and for your willingness to update the score. We greatly appreciate your thorough review and valuable feedback throughout this process.

---

### Author Response · Authors · 2023-11-20
**The rebuttal window is closing soon**

Dear Reviewers,

We truly appreciate your careful feedbacks and constructive suggestions. We believe we have addressed most of your concerns which have subsequently enriched our submission and made our conclusions stronger. Since the rebuttal window is closing soon, we would like to invite you to carefully review our responses, the revised version of the paper & the Appendices containing the new experiments and the evaluation studies requested by you, and finally the subjective demo website attached. We hope these new additions and comments are convincing enough to evaluate our work in a more positive light.

Sincerely,

The authors

---

### Meta-Review · Area_Chair_v2H8 · 2023-12-07

**Metareview:**

The paper presents a language-conditional sound source separation framework able to separate single sources from mixtures without using single-source audio data during training. The proposed method is novel and shows significant improvement over the baselines. The paper is well written.

I am recommending acceptance.

**Justification For Why Not Higher Score:**

This is a very interesting yet not ground-breaking work.

**Justification For Why Not Lower Score:**

The paper received unanimous acceptance from 3 confident reviewers with scores 668.

---

### Decision · Program_Chairs · 2024-01-16

Accept (poster)